

# Physically-Based Modelling of co-seismic Landslide, Debris Flow and Flood Cascade

Bastian van den Bout[1], Chenxiao Tang[2], Cees van Westen[1], Victor Jetten[1]

[1]Faculty of Geo-Information Science and Earth Observation (ITC), University of Twente.

[2]Institute of Mountain Hazards and Environment, Chinese Academy of Sciences & Ministry of Water Conservancy.

*Correspondence to*: B. van den Bout (b.vandenbout@utwente.nl)

**Abstract.** The 2008 Wenchuan earthquake lead to various complex multi-hazard chains that included seismically-triggered landslide initiation, landslide run-out, river damming, dam breaching and flooding. The modelling of the interactions between

such hazardous processes is challenging due to the complexity and uncertainty. Here we present an event-based physically-based model that is able to simulate multi-hazard land surface process chains within a single unified simulation. The final model is used to simulate a multi-hazard chain event in the Hongchun watershed, where co-seismic landslides led to a landslide dam and, two years later, a debris flow that breached the landslide dam. While most aspects of the multi-hazard chain are well predicted, the correct prediction of slope failures remains the biggest challenge. Although the results should be treated

carefully, the development of such a model provides a significant progress in the applicability of multi-hazard chain simulations.

## 1 Introduction

Many of the damaging events that involve land surface processes are not caused by individual but multiple interacting

hazardous processes. Such combinations can take place because the same triggering events (e.g. extreme rainfall) triggers various hazardous processes (e.g. flashfloods, landslides and debris flows) that interact and that may impact the same elements-at-risk. For multi-hazard assessment, the hazard intensities and impact can differ significantly when compared to the individual hazardous processes (Gill & Malamud, 2014; Van Westen and Greiving, 2017). Hazardous events may also occur in sequence as cascading or domino events whereby one hazardous process triggers another either directly or later in time. One particular



hazardous example of such cascading hazard events is the natural damming of rivers by landslides (Costa & Schuster, 1988; Walder & O'Conner, 1997). These landslide dams occur in mountainous landslide-prone areas around the world (Swanson et al., 1986; Chai et al., 2000; Dai et al. 2005; Korup, 2005; Harp and Crone, 2006; Nash et al., 2008; Liu et al., 2009; Fan et al., 2014). This sequence of hazardous events starts with a catastrophic slope failure, often caused by intense precipitation, a seismic trigger, or a combination of these. The released material moves downslope and can lose momentum through friction

or by entering a sharply incised channel or river. When the volume of the solid materials is large enough, or when the moved landslide body retains mostly its original strength characteristics, it will form a barrier for water flow which will accumulate to form a dammed lake. Depending on the strength of the materials composing the dam, the barrier may breach when the barrier lake level exceeds the height of the dam, either due to accelerated erosion, piping or barrier collapse (Ermini and Casagli, 2003). During the breach, extreme discharges and solid-laden floods with high velocity may devastate the downstream

areas (Schuster, 1993; Walder and O'Connor, 1997).

Natural dynamical systems such as the one described above can be complicated, containing many interactions and numerous fundamental processes related to hydrology and sediments (Walder & O'Conner, 1997). Slope failures, mass movement runout and flooding are influenced by catchment scale hydrology (van den Bout et al., 2018). Furthermore, inter-hazard interaction exist in many varieties, a review of which can be found in Kappes et al. (2012). In the case of landslide dam formation and

breaching, many interactions exist between processes that are typically approached individually in modelling. Landslide dam break floods have been analyzed using both empirical and physically-based models (Evans, 1986; Costa and Schuster, 1988; Peng and Zhang, 2012). Empirical models are simpler to apply but provide less comprehensive results than physically-based models. On the other hand, physically based models require detailed physical parameters as input, and can be computationally costly. Whereas the individual components of the hazard chain: landslide initiation, landslide run-out, dam breach and flooding,

have been modelled using physically-based models, the interactions between these processes are generally not simulated within a single model due to their high complexity.

The prediction of landslide volumes resulting from earthquakes is a complex problem, and requires specialized numerical models. Several physically-based simulation tools for slope failure volume modelling have been developd, such as CLARA (Hungr et al., 1989), TSLOPE3 (Pyke, 1991), 3D-SLOPE (Lam and Fredlund, 1993), 3-DSLOPEGIS (Xie et al., 2003),


r.slope.stability (Mergili et al., 2014), Scoops3D (Reid et al., 2015), EDDA (Chen & Zhang, 2015) and OpenLISEM (Van den Bout et al., 2018). CLARA, TSLOPE3, and 3D-SLOPE can only be applied on individual slopes, while r.slope.stability, Scoops3D, and OpenLISEM are spatially distributed models, which are based on Geographic Information Systems (GIS). These can be applied for landslide volume estimation over a large area up to several hundred km$^2$. Numerical modelling of mass movement run-out using 2-D approaches has been implemented in a variety of models (Malet et al., 2004; Rickenmann

et al., 2006; Van Asch et al., 2007; Hürlimann et al., 2007; Domenech et al., 2019). They need detailed information on initial volume, rheology, entrainment and an accurate and detailed digital elevation model (DEM) (Hürlimann et al 2007). Erosion, the water-driven uptake of sediment, and entrainment, the grain-driven uptake of sediment, have been used in understanding mass flow soil interactions. Erosion models, while traditionally focused on agricultural processes, come in great variety and provide insight into the flow-surface interactions. Examples are WEPP (Nearing et al., 1989), EUROSEM (Morgan et al.,

1998) and Delft3D Sediment (Roelvink and Banning, 1995).

In the case of multi-hazard chains including landslide dams, integrated simulations face several critical issues. When a mass movement enters a channel, with a certain water level, the landslide material mixes with the water in a dynamic manner, which is generally ignored in existing flow models such as Flo-2D (O'Brien et al., 1993). Similarly, the volumetric sediment content of water increases when a landslide dam is breached, and the material of the dam is entrained by the water flow. The

entrainment of bed material is simulated in a limited number of spatial two-dimensional mass movement models, but is rarely initiated from low-concentration water flow. Several models have shown this functionality, but lack the capability of modelling breaching behavior, and ignore the resulting changes in the digital elevation model (Chen & Zhang, 2015; Hu et al.,2016). Ignoring these changes makes the simulation of any breaching behavior impossible, since increasing outflow must be the result of the entrainment of a flow path on the landslide dam.

Simplified coupled model approaches have been tested using separate, and non-spatially distributed models for water flow and dam breaching. Empirical equations for dam-breach discharge have been developed and implemented by Singh & Snorasson (1984), Wang et al. (2008) and in the BREACH model (Fread, 1988). In these models, mathematical expressions for the dynamics of the outgoing discharge during a dam breach are derived from simplified landslide dam examples. Typically, a feedback loop between outflowing discharge and the amount of material entrained from the landslide dam determines the
dynamics of the hydrograph. While these provide a useful estimation of the relevant physical processes during a dam breach,

only outgoing discharge is simulated and downstream processes are unknown. Moreover, the accuracy is generally low for

more complicated cases (Zhu, 2006). The BREACH model simulates the increasing breach depth in a landslide dam using an

iterative numerical solution. At the sides of the entrained channel, a limiting angle determines the additional collapse of

material. Valiani et al. (2002) improved this by simulating dam breach discharge using a two-dimensional finite element

method.

Fan et al. (2014) provided an insightful step towards integrated modelling by linking an one-dimensional breach outflow model

with a hydraulic 2D flood simulation. The outflow from the BREACH model determines the boundary condition for the flood

model (Sobek). With this combined setup, it was possible to predict the dynamic dam breaching and the resulting large scale

flood behavior, with significant accuracy. However, this integrated setup was still limited by the assumptions in the model.

The breach model is one-dimensional and uses a simplified shape for the estimation of breaching dynamics. The setup ignores

catchment-scale hydrological processes that could influence the surface flow. Furthermore, breach outflow can typically

contain large amounts of solid material, altering the dynamics of the mixture. Fan et al. (2014) implemented a flood model

where flow is calculated using the Saint Venant equations for shallow flow, which ignores forces such as viscosity, and

implements a fluid-based frictional model (Dhondia and Stelling, 2002). Li et al. (2011) provided a different approach to

integrated simulations of landslide dams by linking the BREACH model with both a regional rainfall-runoff model and the

Flo-2D debris flow model for modelling the runout of the breach material. Despite their improved method, the landslide dam

brech modelling depends on assumptions such as constant flow material properties, a limiting region for entrainment and

landslide initiation coming from pre-defined boundary conditions. More recently, Mergili et al (2017) show the application of

diluting mass flows to modelling Glacial Lake Outburst floods. They developed the r.avaflow model, based on two-phase

mixture equations by Pudasaini (2012) that allows for mixed flow of water and solids and implements a simplified entrainment

process. However, this work does not take into account catchment-scale processes or actual landslide dam formation.

In this research, we aim to simulate a complex multi-hazard chain using a physically-based integrated model. We present the

implementation of a complete spatial simulation of a landslide dam process chain, including initial slope failure, landslide

runout, deposition, lake formation, dam breaching and flooding. To test the behavior of the developed model, simulations and

validation will be shown for a case study of a dam-break flood event in the Hongchun watershed, that occurred in 2010 near

Yinxiu town, close to the epicenter of the 2008 Wenchuan earthquake, in Sichuan Province, China. Finally, we investigate the

predictive capabilities of complex multi-hazard multi-stage simulation by analyzing the sensitivity of the model to changes in

input parameters.

The investigation of the Hongchun watershed builds on previous works in literature. Tang et al. (2011; 2015) describe the co-

seismic and post seismic landslide events between 2008 and 2011 in this area. Several other studies have simulated the event

using a variety of modelling techniques. Ouyang et al. (2015) applied shallow flow depth-averaged debris flow equations in

order to understand the event as a simplified runout process. Zhang et al., (2018) utilized a depth-averaged smooth particle

hydrodynamics model to simulate both runout from landslides and the later debris flow. The authors show a novel application

of such methods to a multi-stage event, however, without an integration in catchment scale hydrology and a physical

implementation of entrainment and breaching of the landslide dam. Domenech et al. (2019) used a multi-event debris flow

model including entrainment to study the effect of material depletion on debris flow initiation in the Hongchun watershed.

Using results of modelling studies, Chen et al. (2016) performed a cost-benefit analysis for the mitigation measures to protect

the touristic town of Yinxiu, located directly opposite to the outlet of the Hongchun watershed on the other side of the Ming

River.

In this research, there is a focus on the integrated nature of the developed multi-hazard model. Multi-hazard multi-stage

behavior is simulated for a series of interacting earthquake, landslide, debris flow and flood processes in the Hongchun

watershed. In order to simulate the behavior of this complex event, we develop an extended and improved version of

OpenLISEM hazard. To analyze the uncertainties in modelling such process chains, we employ ensemble simulations and

analyze spatial hazard probabilities to estimate reliability. Finally, we discuss the benefits, downsides and potential application

of modelling methods that involve integrated multi-hazard process chains.



## 2 Theoretical Model Background

This section presents the theoretical background and governing equations for the components of our multi-hazard model. One of the important cornerstones is the work by Pudasaini (2012) who developed a set of physically-based two-phase mass
movement equations that can adapt the internal forces in the flow based on the local volumetric solid concentration. This allows to simulate the behavior of landslides, the flow of water and the interactions between mass movements and water flow (Mergili et al., 2018). Using these equations, van den Bout et al. (2018) developed an integrated model for slope hydrology, slope failure, mass movements and runout.

We have implemented a series of new processes within the physically-based OpenLISEM Hazard model. In this section, we
describe relevant theory of existing functionality, as well as the addition of terrain-altering entrainment by mixture flows, and the simulation of co-seismic shallow landslides. An overview of the processes is provided in figure 1.

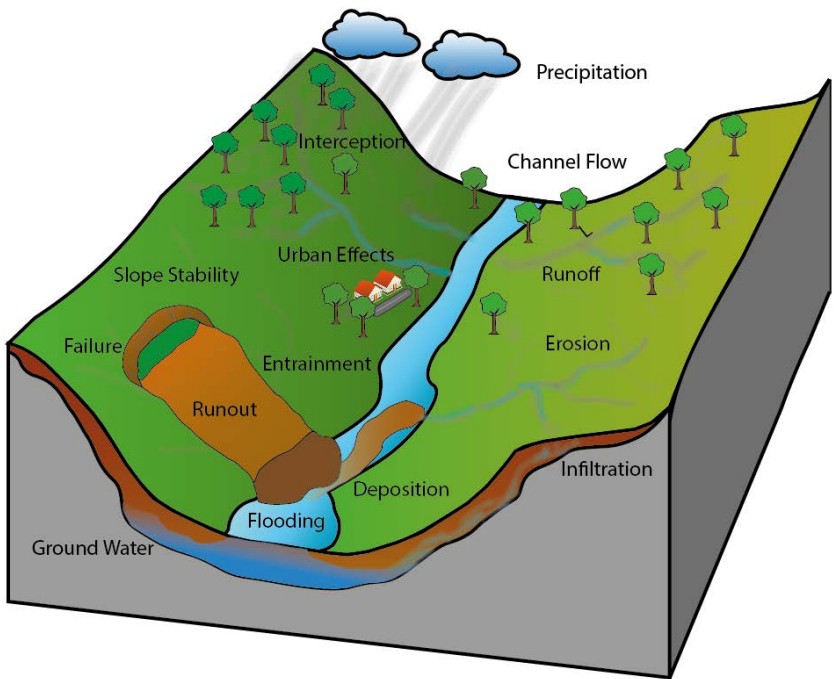

Figure 1 A schematic overview of processes, fluxes and storages within the OpenLISEM model.

**1.1 Hydrology**



The principle setup of the model starts with the surface aspects of the hydrological cycle. Spatially and temporally distributed

rainfall forms the main input of water. We implement a set of equations for interception and micro-surface storage based on

the original version of OpenLISEM (Jetten & de Roo, 2004; Baartmans et al., 2013). The Green & Ampt infiltration model is

implemented, which assumes a wetting front moving down into the soil due to infiltrating rainfall (Green & Ampt, 1911). The

resulting potential infiltration is subtracted from the available surface water (Equation 1).

$$f_{pot} = -K_s \left( \psi \, \frac{\theta_s - \theta_i}{F} + 1 \right)$$

Where $f_{pot}$ is the potential infiltration rate ($m\ s^{-1}$), $F$ is the cumulative infiltrated water ($m$), $\theta_s$ is the porosity

($m^3\ m^{-3}$), $\theta_i$ is the initial soil moisture content ($m^3\ m^{-3}$), $\psi$ is the matric pressure at the wetting front ($h = \psi + Z$)

($m$) and $K_s$ is the saturated conductivity ($m\ s^{-1}$).

We simulate groundwater flow using depth-averaged Darcy equations. A separate modelling component is used to simulate

several months of groundwater dynamics and generate the initial ground water conditions for the OpenLISEM Hazard

simulation, which silumates the effects of a single rainstorm. Groundwater flow is estimated using a depth-averaged Darcy

equation (Equation 2), as is common in regional hydrological models (Van Beek, 2002).

$$V_d = K_s * \frac{dh}{dx}$$

Where $h$ is the hydraulic head (m) and $V_d$ is the darcy flow velocity ($m\ s^{-1}$)

### 1.2 Slope Stability and Failure

Hydrology influences slope stability and can eventually lead to failure. Slope failure is based on the Iterative Failure Method

(Bout et al., 2018). This technique reverses the Factor of Safety (equation 3) to solve for the remaining depth of material at

which the local situation becomes stable. The locally altered terrain then results in changed forces in the surrounding cells.

Through iteration, the method keeps removing material until no unstable cell is left and the minimum required material for a

stable terrain has been removed. We added the seismic forcing in the Factor of Safety calculation following Morgenstern &

Sangrey (1978).

$$FOS = \frac{c' + \left( \left( (\gamma - m\gamma_w) h_s + m\gamma_w z \right) \cos(\beta)^2 - h_s \gamma \alpha \sin(\beta) \cos(\beta) \right) \tan(\phi)}{\left( (\gamma - m\gamma_w) h_s \right) \sin(\beta) \cos(\beta) + h_s \gamma \alpha \cos(\beta)^2}$$


Where $FOS$ is the Factor of Safety (-), $\beta$ is the slope of the soil section (-), $c'$ is the apparent cohesion of the soil (kPa), $\alpha$ is the

peak horizontal earthquake acceleration ($m\ s^{-2}$), $\phi$ is the internal friction angle of the soil (-), $\gamma$ is the density of the slope

material ($kg\ m^{-3}$), $\gamma_w$ is the density of water ($kg\ m^{-3}$), m is the fraction of the soil depth that is saturated from the basal

boundary (-) and $h_s$ is the depth of the failure plane ($m$).

The apparent cohesion consists of additional root cohesion and a matric suction term. The acceleration is assumed to be, as in

the most critical situation, the estimated peak horizontal ground acceleration. To account for sub-surface force propagation,

we include an iterative force solution. This is an extension of similar approaches in Zhou & Cheng (2013) and Zhou et al.,

(2014). In the proposed implementation, forces are iteratively solved throughout the entire terrain description (Figure 2).

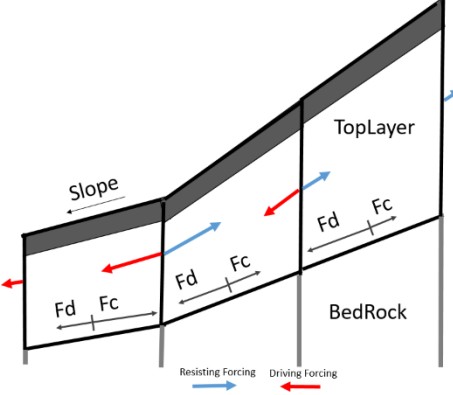


Figure 2 Sub-surface force distribution is solved through iteratively finding a steady state (Fd = driving force, Fc = resisting

force)

In the three-dimensional case, where the x-and y- components of the seismic forcing and slope steepness influence the

propagation, this can be expressed as equation 4.


4 $$\nabla \overrightarrow{F_{up}} + \left(\vec{C} - \vec{D}\right) * \left(\vec{S} \cdot \overrightarrow{F_{lat}}\right) = 0$$





With $C$ the force capacity (numerator of equation 3), D the force demand (denominator of equation 3), $\vec{S}$ is the normalized

slope vector (-) and $\overrightarrow{F_{lat}}$ is the vector of laterally acting forces ($kg\ m\ s^{-2}$).

We assume that excess force is transferred downslope, but the fractured top material is unable to transfer force resistance

upslope.

We assume in the set-up of the model that a failure plane can develop in the soil materials at any depth. Equation 3 can be

inverted to find the value of h, for which the safety factor is one. To solve this equation for h, we first express the slope based

on the local elevation differences and create a shortened equation for the factor of safety (equation 5 and 6).

5
$$\beta = atan\left(\frac{\max(z_{x-1}-z_x, z_x-z_{x+1})}{dx}\right)$$

$$FOS = 1 = \frac{C_1 + C_2 \cdot cos\left(atan\left(\frac{z-z_0}{dx}\right)\right)^2 \cdot \tan(\phi)}{C_3 \cdot \sin\left(atan\left(\frac{z-z_0}{dx}\right)\right) \cdot cos\left(atan\left(\frac{z-z_0}{dx}\right)\right)}$$

Where the simplified constants are given by equations $C_1 = c, C_2 = ((\gamma - m\gamma_w)h_s + m\gamma_w h_s), C_3 = ((\gamma - m\gamma_w)h_s), \beta$ is

the slope angle (-), $z$ is the elevation above the failure surface (m) and $z_0$ is the lowest neighboring elevation (m).

Solving this equation can be done using the trigonometric identities (Equation 7 and 8).

7
$$cos(atan(x)) = \frac{1}{\sqrt{1+x^2}}$$

$$sin(atan(x)) = \frac{x}{\sqrt{1+x^2}}$$

Finally, we find the lowest real root to the second-order polynomial equation of the form

$$h_s = a + (b)\,x + (c)x^2$$

Where

10
$$a = C_1\,h_0^2 - C_1\,dx^2$$

$$b = 2\,C_1 h0 - C_3\,h_0\,FOS - C_3 h_0 FOS\,dx\,C_2 dx^2$$

$$c = C_1 - C_3\,FOS\,dx$$

Using equation 9 the critical depth can be found, where the material on the slope is in equilibrium. By multiplying the area of

a pixel with the slope material height above this critical depth, a failure volume can be calculated. This volume consists of


solids and water, depending on the soil saturation level, and is then added to the flow equations that simulate Mohr-Coulomb

mixture flow.

### 1.3  Flow Dynamics

Once a volume of flowing material is introduced (either through hydrology or slope failure), its movement must be described.

The dynamics of mass and momentum are calculated using a discretization of the continuity equation for mass and momentum

of both solids and fluids (Equation 13, 14 and 15). Equation 13 described the conservation of mass, and equation 14 describes

conservation of momentum. On the left hand side the storage term and advective terms are found. External forces and mass

sources are added on the right hand side.

$$\frac{\partial h}{\partial t} + \frac{\partial(hu_x)}{\partial x} + \frac{\partial(hu_y)}{\partial y} = R - I$$

$$\frac{\partial hu_x}{\partial t} + \frac{\partial(hu_x^2)}{\partial x} + \frac{\partial(hu_x u_y)}{\partial y} = gh(S_x - S_{f,x})$$

**15**
$$\frac{\partial hu_y}{\partial t} + \frac{\partial(hu_y^2)}{\partial y} + \frac{\partial(hu_x u_y)}{\partial x} = gh(S_y - S_{f,y})$$

With h being the flow height $(m)$, u the flow velocity $(m\ s^{-1})$, R the rainfall $(m)$, I the infiltration $(m)$, g the gravitational

acceleration $(m\ s^{-2})$, S the friction term $(m\ s^{-2})$ and $S_f$ the momentum source term $(m\ s^{-2})$.

The momentum source terms predominantly determine the behavior of the flow. Here we implement pressure and friction

forces for fluids as is typical for shallow water assumptions (O'Brien et al., 1993).

215  16
$$S_x = -\frac{d\left(\frac{gh^2}{2}\right)}{dx} - ghS_{f\ x}$$

$$S_y = -\frac{d\left(\frac{gh^2}{2}\right)}{dy} - ghS_{f\ y}$$

The friction slope terms, which are the friction forces divided by the water height and the gravitational acceleration, can be

calculated using the Darcy-Weisbach friction law (Chow, 1959) (Equation 16).

$$S_f = \frac{g}{n^2}\frac{\vec{u}|\vec{u}|}{h^{\frac{4}{3}}}$$

With n the Manning's n friction coefficient $(s\ m^{-\frac{1}{3}})$.


The momentum source terms within OpenLISEM Hazard are based on the work by Pudasaini (2012). This set of equations contains a physically-based two-phase momentum balance. Besides pressure and gravitational forces, it includes viscous forces, non-Newtonian viscosity, two-phase drag and a Mohr-Coulomb type friction force for the solid phase (Equations 19, 20, 21, 22). Based on the current and local state of flow, forces change in magnitude. This approach allows for a smooth transition between non viscous flow, hyperconcentrated streamflow, debris flows or landslide runout by automatically scaling and solving the interactions between solids and fluids.

$$S_{x,s} = \alpha_s \left( g \left( \frac{\partial b}{\partial x} \right) - \frac{u_s}{|\vec{u}_s|} \tan(\partial P_{b_s}) - \varepsilon P_{b_s} \left( \frac{\partial b}{\partial x} \right) \right) - \varepsilon \alpha_s \gamma P_{b_f} \left( \frac{\partial h}{\partial x} + \frac{\partial b}{\partial x} \right) + C_{DG}(u_f - u_s)|\vec{u}_f - \vec{u}_s|^{j-1}$$

$$S_{y,s} = \alpha_s \left( g \left( \frac{\partial b}{\partial y} \right) - \frac{v_s}{|\vec{u}_s|} \tan(\partial P_{b_s}) - \varepsilon P_{b_s} \left( \frac{\partial b}{\partial y} \right) \right) - \varepsilon \alpha_s \gamma P_{b_f} \left( \frac{\partial h}{\partial y} + \frac{\partial b}{\partial y} \right) + C_{DG}(v_f - v_s)|\vec{u}_f - \vec{u}_s|^{j-1}$$


$$S_{x,s} = \alpha_f \left\{ g \left( \frac{\partial b}{\partial x} \right) - \varepsilon \left[ \frac{1}{h} \frac{\partial}{\partial x} \left( \frac{h^2}{2} P_{b_f} \right) + P_{b_f} \frac{\partial b}{\partial x} - \frac{1}{\alpha_f N_R} \left( 2 \frac{\partial^2 u_f}{\partial x^2} + \frac{\partial^2 v_f}{\partial y \partial x} + \frac{\partial^2 u_f}{\partial y^2} - \frac{\chi u_f}{\varepsilon^2 h^2} \right) + \frac{1}{\alpha_f N_R} \left( 2 \frac{\partial}{\partial x} \left( \frac{\partial \alpha_s}{\partial x} (u_f - u_s) \right) + \right. \right. \right.$$
$$\frac{\partial}{\partial y} \left( \frac{\partial \alpha_s}{\partial x} (v_f - v_s) + \frac{\partial \alpha_s}{\partial y} (u_f - u_s) \right) \right) - \frac{\xi \alpha_s (v_f - v_s)}{\varepsilon^2 \alpha_f N_{RA} h^2} \right\} - \frac{1}{\gamma} C_{DG}(u_f - u_s)|\vec{u}_f - \vec{u}_s|^{j-1}$$

$$S_{y,s} = \alpha_f \left\{ g \left( \frac{\partial b}{\partial y} \right) - \varepsilon \left[ \frac{1}{h} \frac{\partial}{\partial y} \left( \frac{h^2}{2} P_{b_f} \right) + P_{b_f} \frac{\partial b}{\partial y} - \frac{1}{\alpha_f N_R} \left( 2 \frac{\partial^2 v_f}{\partial y^2} + \frac{\partial^2 u_f}{\partial y \partial x} + \frac{\partial^2 v_f}{\partial y^2} - \frac{\chi v_f}{\varepsilon^2 h^2} \right) + \frac{1}{\alpha_f N_R} \left( 2 \frac{\partial}{\partial y} \left( \frac{\partial \alpha_s}{\partial y} (v_f - v_s) \right) + \right. \right. \right.$$
$$\frac{\partial}{\partial y} \left( \frac{\partial \alpha_s}{\partial y} (u_f - u_s) + \frac{\partial \alpha_s}{\partial x} (v_f - v_s) \right) \right) - \frac{\xi \alpha_s (u_f - u_s)}{\varepsilon^2 \alpha_f N_{RA} h^2} \right\} - \frac{1}{\gamma} C_{DG}(u_f - u_s)|\vec{u}_f - \vec{u}_s|^{j-1}$$


With $S_s$ is the momentum source term for solids $(m\ s^{-2})$ $S_f$ is the momentum source term for solids (), $\alpha_s$ and $\alpha_f$ the volume fraction for solid and fluid phases (-), $P_b$ the pressure at the base surface $(Kg\ m^{-1}s^{-2})$,$b$ the basal surface of the flow $(m)$, $N_R$ the Reynolds number (-), $N_{RA}$ the quasi-Reynolds number (-), $C_{DG}$ the drag coefficient (-), $\rho_f$ is the density of the fluid $(kg\ m^{-3})$, $\rho_s$ is the density of the solids $(kg\ m^{-3})$, $\gamma$ the density ratio between the fluid and solid phase(-), $\chi$ the vertical shearing of fluid velocity $(m\ s^{-1})$, $\varepsilon$ the aspect ratio of the model(-), $\xi$ the vertical distribution of $\alpha_s$ $(m^{-1})$.





Our model follows the equations by Pudasaini (2012) for the definitions of the regular and interface Reynolds number, which represents the ratio between inertial and viscous forces within the fluid (Equation 23 and 24).

$$N_R = \frac{\sqrt{gLH}\,\rho_f}{\alpha_f \eta}$$

245    24
$$N_{R_A} = \frac{\sqrt{gLH}\,\rho_f}{A\eta}$$

Where $L$ is the length scale of the flow (m), $H$ is the height of the flow (m), $\eta$ is the viscosity ($kg\ s^{-1}\ m^{-1}$) and $A$ is the mobility of the interface, scaling parameter for non-Newtonian viscous-fluid stresses ($\approx 1 - \alpha_s$) (-).

To apply these two-phase equations successfully in a catchment-based model, we replace the frictional force for the water

phase with the Darcy-Wiesbach equation for water flow friction. To complete the set of equations that govern debris flow-dynamics, several flow properties were estimated based on the volumetric sediment content. Viscosity is based on an empirical relation by O'Brien and Julien (1985) (Equation 25).

$$\eta = \alpha e^{\beta \alpha_s}$$

Where $\alpha_s$ is the volumetric solid content of the flow (-), $\alpha$ is the first viscosity parameter (-) and $\beta$ the second viscosity

parameter (-).

The drag coefficient is based on the relation provided by Pudasaini (2012) (Equation 26, 27, 28 and 29).

$$C_{DG} = \frac{\alpha_f \alpha_s \left(1 - \frac{\rho_f}{\rho_s}\right)}{\varepsilon U_T (PF(Re) + (1-P)G(Re))}$$

$$Re = \frac{\rho_f d U_T}{\eta_f}$$

$$F = \frac{\rho_f}{180\,\rho_s} \left(\frac{\alpha_f}{\alpha_s}\right)^3 Re$$

**29**
$$G = \alpha_f^{M(Re)-1}$$



Where F and G are the scalar functions describing the flow velocity of solids and fluids respectively (-), Re is the particle

Reynolds number (-),d is the median grain diameter (-), $U_T$ the settling velocity $(m\,s^{-1})$ and M is an empirical parameter

depending on the Reynolds number $(\approx 0.2)$ (-).

Finally, the settling velocity of small $(d < 100\,\mu m)$ grains is estimated by Stokes equations for a homogeneous sphere in water

(Stokes, 1850). For larger grains $(> 1mm)$, the equation by Zanke (1977) is used (Equation 30).

$$30 \qquad U_T = 10\,\frac{\frac{\eta^2}{\rho_f}}{d}\left(\sqrt{1 + \frac{0.01\left(\frac{(\rho_s - \rho_f)}{\rho_f}gd^3\right)}{\frac{\eta}{\rho_f}}} - 1\right)$$

In which $U_T$ is the settling (or terminal) velocity of a solid grain $(m\,s^{-1})$, $\eta$ is the dynamic viscosity of the fluid (), $\rho_f$ is the

density of the fluid $(kg\,m^{-3})$, $\rho_s$ is the density of the solids $(kg\,m^{-3})$, d is the grain diameter $(m)$

## 1.4 Deposition and Dam-Formation

Flows such as the ones described by the equations presented above have complex interactions with the basal surface. In

particular, material in the flow can be deposited, or material from the terrain can be entrained. Deposition occurs when the

flow velocities of a solid-fluid mixture have sufficiently low velocities and drag forces are insignificant compared to shear

resistance. Then, water and solids are subtracted from flow volumes to form a saturated deposits layer. Further deposition

occurs as an active process, during flow, based on the deposition equations of Takahashi (1992). These equations use local

stability analysis and the ratio between the flow velocity and the critical velocity to estimate deposition of solids (Equation 31,

32, 33 and 34). Using these generalized deposition equations, a variety of more specific deposition-based processes can be

potentially simulated. For example, deposition of landslide material in a river would equate to landslide dam formation in

rivers.

$$31 \qquad D = \left(1 - \frac{|\vec{u}|}{p|\vec{u}|_{cr}}\right)\frac{\alpha_{eq} - \alpha}{\alpha^b}V$$

$$32 \qquad |\vec{u}|_{cr} = \frac{\frac{2}{5d_{50}}\sqrt{\frac{g\sin(\theta_c)\rho}{0.02\rho_s}}1}{\left(\frac{\alpha^b}{\alpha}\right)^{-\frac{1}{3}} - 1}h^{1.5}$$





$$\tan(\theta_c) = \frac{\alpha(\rho_s - \rho_w)\tan(\phi^b)}{\alpha(\rho_s - \rho_w) + \rho_w}$$

$$\alpha_{eq} = \frac{\rho_w \tan(\theta)}{(\rho_s - \rho_w)(\tan(\phi^b) - \tan(\theta))}$$

Where D is the deposition rate ($m\ s^{-1}$), $|\vec{u}|_{cr}$ is the critical velocity for deposition ($m\ s^{-1}$), $p$ is the calibration factor for the critical velocity for deposition (-), $\alpha_{eq}$ is the equilibrium volumetric solid concentration (-), $\alpha^b$ is the volumetric solid concentration of the bed material (-), and $d_{50}$ is the median grain size ($m$).

### 1.5 Entrainment Equations

In order to estimate entrainment, we implement the equations by Takahashi et al. (1992) in a similar manner as was done in the Edda model (Chen and Zhang, 2015). The expressions for the entrainment rate are provided by equation 35, 36 and 37.

$$E = K(\tau - \tau_c)$$

$$\tau = \rho g h S_f$$

$$\tau_c = c^b + (1 - C_s)\alpha(\rho_s - \rho_w)gh\ cos(\theta)^2 \tan(\phi^b)$$

Where E Is the rate of change of the basal topography (erosion rate) ($ms^{-1}$); $\tau$ is the shear stress (Pa), $\tau_c$ is the critical shear stress (Pa), $S_f$ is the surface friction term (-), $\tau_y$ is the yield stress (Pa), $K$ is the resistance parameter for laminar flow (-), $n_{td}$ is the turbulent dispersive coefficient ($m^{\frac{1}{2}}s^{-1}$), $c^b$ is the cohesion of the bed material (Pa) and $C_s$ is the coefficient of suspension (-).

The surface shear term is calculated from the momentum conservation equation as the sum of all surface frictional terms. We adapt the expressions to conform to the momentum jump boundary condition, as provided by Iverson and Ouyang (2015) (Equation 38).

$$\frac{dz}{dt} = E = \frac{\tau - \tau_c}{\rho_{eff} u_b}$$

Where $\rho_{eff}$ is the total effective density of the flow ($kg\ m^3$), $u_b$ is the basal velocity ($m\ s^{-1}$).

Iverson and Ouyang indicate that the equations by Takahashi do not include a distinction between basal and mean velocities and do not conserve momentum. The basal velocity differs from the mean velocity of the flow according to a vertical velocity profile. Our model description does not include an estimate of the basal flow velocity. Therefore, we use the entrainment




coefficient as a scaling device to convert from the mean velocity to a basal velocity ($u_b \approx K\,\bar{u}$), with $\bar{u}$ the depth-averaged

velocity of the flow. This approach was similarly taken by Pudasaini and Fischer (2016). Therefore,

$$\frac{dz}{dt} = E \approx \frac{\tau - \tau_c}{\rho_{eff}\,K\,\bar{u}}$$

Furthermore, we ignore vertical velocities in the model setup, a both common and necessary assumption that has given good

results in other models (Iverson and Ouyang, 2015). Finally, we alter the mass and momentum source terms to include the

produced mass and momentum from entrainment and thereby hold to conservation of these quantities. We introduce into the

flow, based on the entrainment rates, a new mixture of solids and fluids. The momentum of the flowing material is additionally

increased by adding the momentum of the entrained mass, which gains a velocity equal to the estimated basal velocity. Lateral

entrainment includes collapse due to slope failure which is included by solving for stable depth. This is automatically solved

by the iterative failure method as described previously. Slope failure is based on a local safety factor estimation which is

inverted as is done for slope stability.

### 1.6 Flowchart and numerical implementation

A flowchart of the full model is given in figure 3. Hydrology forms the basis of the simulated cycle. From this, other sediment

and solids-related processes are linked. At each timestep, properties of the flow are determined based on the actual water and

solid content. Furthermore, solid properties such as density, friction angle and size are advected with the solids. The flowcharts

indicates the data required for each step of the simulation. This highlights the downside of integrated modelling approaches,

the increase in required input data.



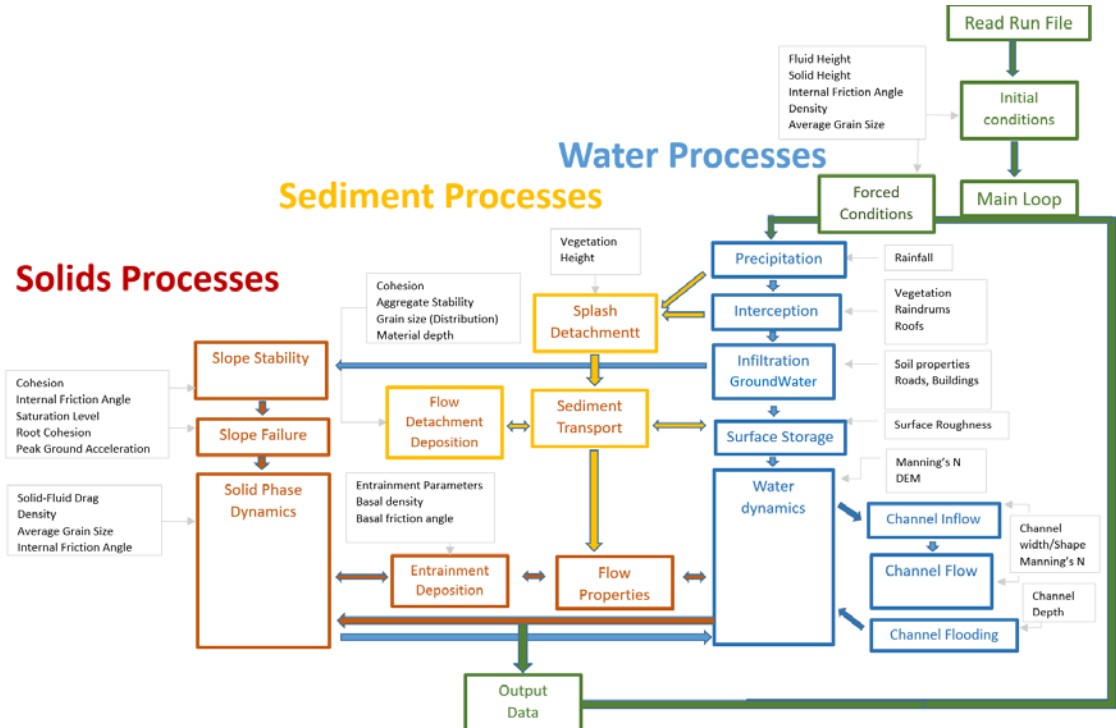

Figure 3 A schematic overview of the OpenLISEM Hazard model, including the link with the most relevant input data.

Similar catchment-integrated simulations that include hydrology, entrainment and debris flow initiation have been used

successfully by the EDDA model (Chen & Zhang, 2015), Hu et al. (2016) and by the STREP TRAMM model (Lehmann and

von Rütte, 2017). These models predict entrainment of slope material or landslide deposits, but do not alter the elevation

model. By using recent developments in numerical schemes, elevation model changes can be simulated without erroneous

feedbacks between terrain and flow. The numerical scheme within the model is based on the monotonic upstream cell-centered

scheme (MUSCL) (Van Leer, 1979). This scheme uses piecewise linear reconstructions of both the terrain surface and the

flow properties. This terrain reconstruction estimates local slope, and in rough terrain, also includes cell boundary elevation

differences. Besides shear stress from the bottom surface in a cell, cell boundary barriers can also provide shear resistance to

the flow. Our implementation of entrainment therefore takes into account direct and lateral entrainment.





## 3 Study Case

The integrated OpenLISEM multi-hazard model was applied in the Hongchun watershed, located near Yingxiu town in the
epicentral area of the 2008 Wenchuan earthquake, in Sichuan province, China. This watershed experienced numerous co-
seismic landslides and a catastrophic debris flow event in 2010. The debris flow deposits from this watershed dammed the
main Min River and flooded Yinxiu town (Figure 4).

This watershed has an area of 5.3 km$^2$, and elevation ranges between 900 and 1700 meters. The very steep slopes of this
watershed (>30 degrees) were covered by dense vegetation before the earthquake in 2008. The steep terrain and sharply incised
channels provide few locations for human settlements. The touristic town of Yinxiu is located adjacent to the Min river, and
opposite of the outlet of Hongchun watershed. The lithology of the area consists mainly of highly fractured granitic rocks, with
some pyroclastic rocks, limestones and sandstones (Tang et al., 2011). The texture of the weathered material is predominantly
clay-loam with large amounts of gravel. The Beichuan thrust fault runs straight through the watershed (Figure 4; Mahodja et
al., 2016). In 2008, the fault was ruptured in the Wenchuan earthquake ($M_w$ 7.9). After the earthquake, numerous landslides
occurred in the area, leaving large volumes of deposits in the streams and channels, and removing the vegetation in 50% of
the area. A detailed co-seismic landslide inventory from Tang et al. (2016) is shown in figure 4. The northern part of the
catchment, which is part of the hanging wall, was more impacted than the southern part. One of the largest landslides in the
watershed blocked the main channel of the Hongchun catchment (Figure 4).


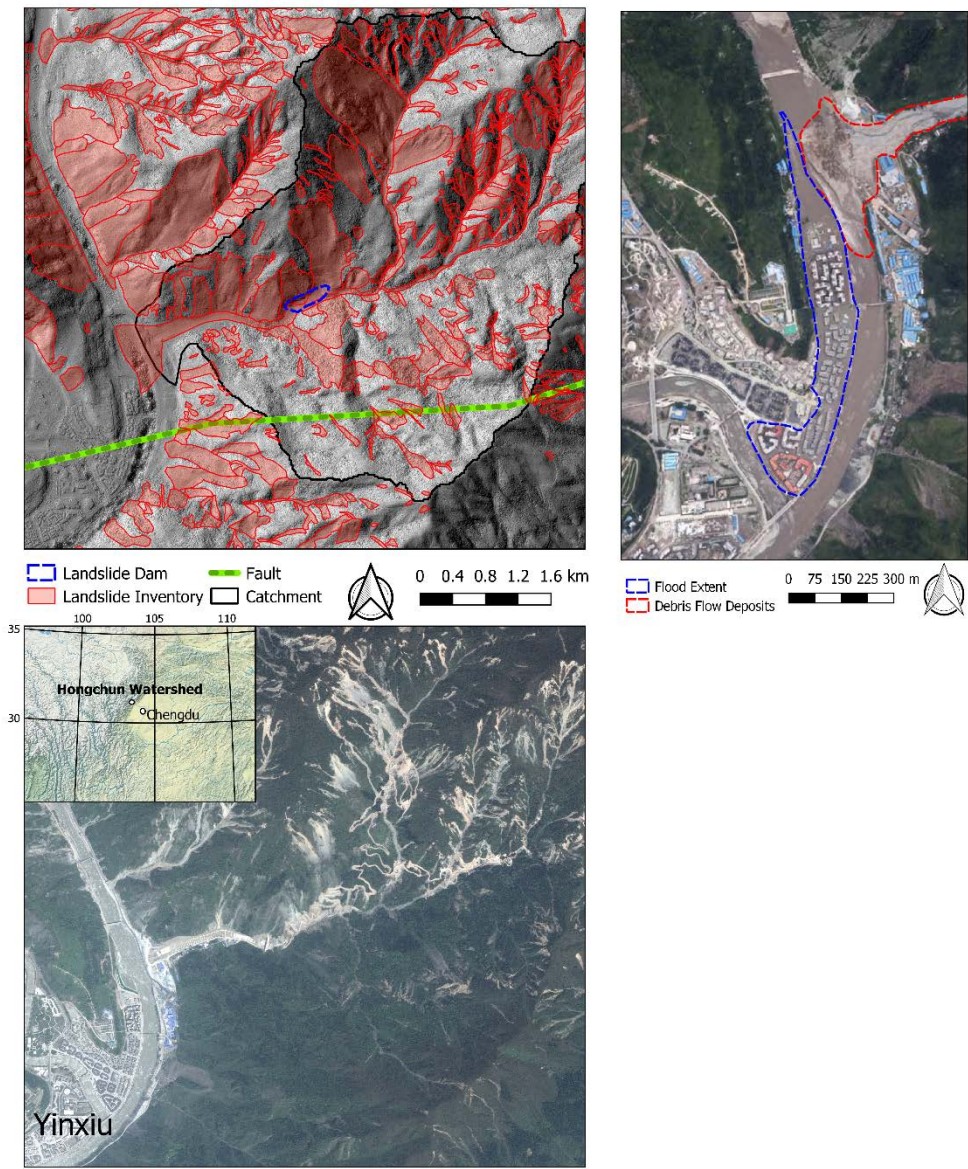

**Figure 4 An overview of the Hongshun Watershed: (Top) Hillshade Image with co-seismic landslide map from Tang et al. (2016) (bottom) Post-Event natural colour composite from Pléiades ssatellite, showing the situation in 2017. The construction of massive debris flow mitigation works in the outlet of the watershed are clearly visible, as is the continued mass movement activity, 9 years after the earthquake. (right) Aerial image of the 2010 debris flow blocking the Min river and causing flooding in Yinxiu town.**

### 3.1 The Multi-Hazard Event of 2010


In 2010, two years after the earthquake the area experienced several rainy weeks followed by a high intensity rainfall event. This event consisted of two peaks of several hours of rainfall, with intensities up to 33 millimeters per hour. In total, 220 mm of rainfall fell in two days.

During this event, a debris flow was generated by entrainment of loose landslide deposits, and the landslide dam located in the central channel was breached. Due to the dam breaching, the volume of the flow increased substantially, also due to more

entrainment downstream. Upon leaving the Hongchun watershed, the debris flow material deposited in the Min River, with a thickness of up to 15 meters high and an estimated total volume of 7.11 $\times 10^5$ $m^3$ (Tang et al., 2011). The Min River, which experienced a high discharge at this moment, was diverted laterally and flooded parts of the nearby newly reconstructed Yinxiu town (Figure 6). For a more detailed description of the event, see also Tang et al. (2011) and Ouyang et al. (2015). A schematic overview of the event is provided in figure 5.

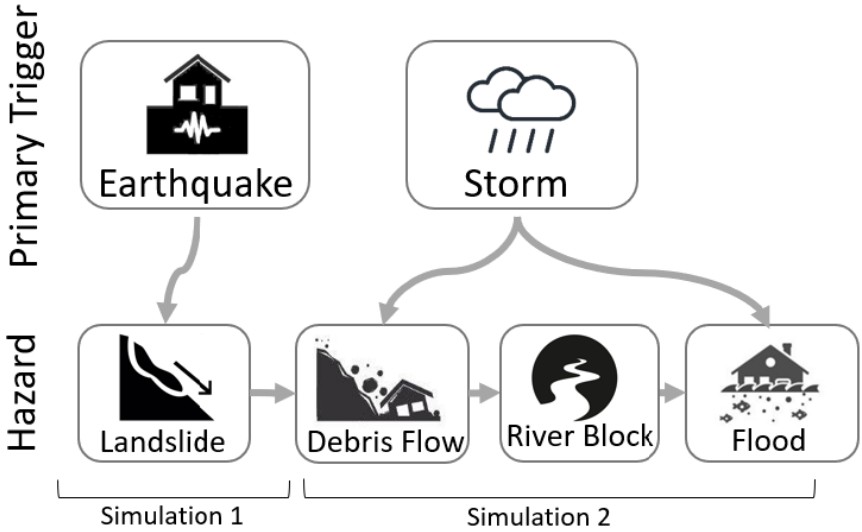


**Figure 5 A schematic overview of the stages of the described event. The events for simulation 1 occurred directly after the Earthquake in 2008. The events for simulation 2 after the rainfall event in 2010.**

In summary, the study area experienced two different multi-hazard chains that will be simulated.

- Co-seismic chain: The first one was experienced during the earthquake, when ground shaking and topographic

amplification triggered a series of co-seismic landslides, some of which blocked the Hongchun stream;



- Post-earthquake chain: heavy rainfall triggered debris flows due to entrainment, which broke the co-seismic landslide dam. The resulting debris flow dammed the Min river, which was diverted into the town of Yingxiu.

All simulations include entrainment of estimated soil depth, hydrology and the related surface processes as described in the theory section.

### 3.2 Model Input and parameters

The input data is based on a combination of laboratory or field measurements and drone or satellite-based spatial data products (Table 1).

*Table 1 List of input data and sources for the multi-stage multi-hazard modelling with OpenLISEM Hazard.*

## Spatial input parameters

| Base Map | Parameter | Source |
|---|---|---|
| Elevation | Pre-Earthquake DTM ($z$) | 20m resolution elevation product from interpolated contourlines |
| | Post-Earthquake DTM ($z$) | Drone photogrammetry 2-meter surface elevation (filtered to DTM using PIX4D vegetation filter) |
| Land Surface | Land cover classes | Sentinel-2 classification at 10m resolution (trained spectral angle mapper) |
| | Mannings N ($n$) | Literature comparison with field photos |
| | NDVI | Landsat-8 images at 30 m resultion (2008) SPOT-4 product at 4m resolution (2010) |
| | Vegetation Cover | Estimated from NDVI using empirical method (Kalacska et al., 2004; Jiang et al., 2006) |

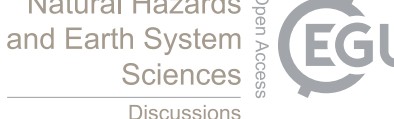

| | | |
|---|---|---|
| | Root Cohesion ($c_r$) | Measured in field (12 samples) and extended based on land cover classes. |
| Soil Material | Texture ($d_{50}$) | Measured from field samples (4 samples) |
| | Saturated Conductivity ($k_{sat}$) | Measured from field samples (4 samples) |
| | Internal Friction Angle ($\phi$) | Measured from field samples (16 samples) Literature values |
| | Cohesion ($c$) | Measured from field samples (16 samples) |
| | Porosity ($\theta_s$), Matric Suction ($\psi$) | Literature Values derived from texture (Saxton et al., 2006) |
| | Density ($\rho$) | Measured from field samples (16 samples) |
| | Initial Moisture ($\theta_i$) | Ground water model run for three months using GPM 30 minute interval satellite precipitation estimates. |
| | Soil Depth ($h_s$) | Soil depth modelling calibrated using landslide scarp depth (method from Ruette et al., 2013) |
| Shakemap | Peak Ground Acceleration ($\theta_s$) | USGS shakemap (Wald et al., 2005) |
| Rainfall | Rainfall Intensities ($R$) | Rainfall station within Yinxiu provided high-accuracy hourly data. Additionally GPM 30-minute interval global rainfall product was used for pre-event ground water modelling. |





| Inventory | Landslide locations | Mapped from high-resolution imagery (Tang et al., 2017) |
|---|---|---|

**Global parameters**

(besides multipliers for all spatial input data with default value of 1.0)

| $\alpha$ | $\beta$ | $K$ | $C$ | $P$ | $C_V$ |
|---|---|---|---|---|---|
| 1 | 10 | 0.05 | 0.1 | 0.5 | 0.65 |

A pre-earthquake digital terrain model with 20 meters spatial resolution was obtained from the local government. Unfortunately, we were not able to obtain better quality pre-earthquake elevation data. Post-earthquake surface elevation data were available at 2-meters spatial resolution, acquired from fixed-wing drone flights (See Figure 6) and filtered using the Pix4D DTM filter to remove vegetation (Pix4D, 2017). Flow simulations typically are most dependent on the accuracy and spatial resolution of the elevation model. In order to have an effective compromise between detail and computation time, we

resampled all base input to maps of 10 meters resolution

Pre-earthquake NDVI values were calculated at 30 meter resolution Landsat-8 images from 2008. A post-earthquake NDVI map was obtained from 4 meter spatial resolution SPOT-images acquired in 2009. NDVI values were used to estimate Leaf Area Index (LAI) by applying an empirical relation obtained from tropical forest data (Kalacska et al., 2004). We used it to estimate fractional vegetation cover using a similar empirical relationships (Jiang et al., 2006).

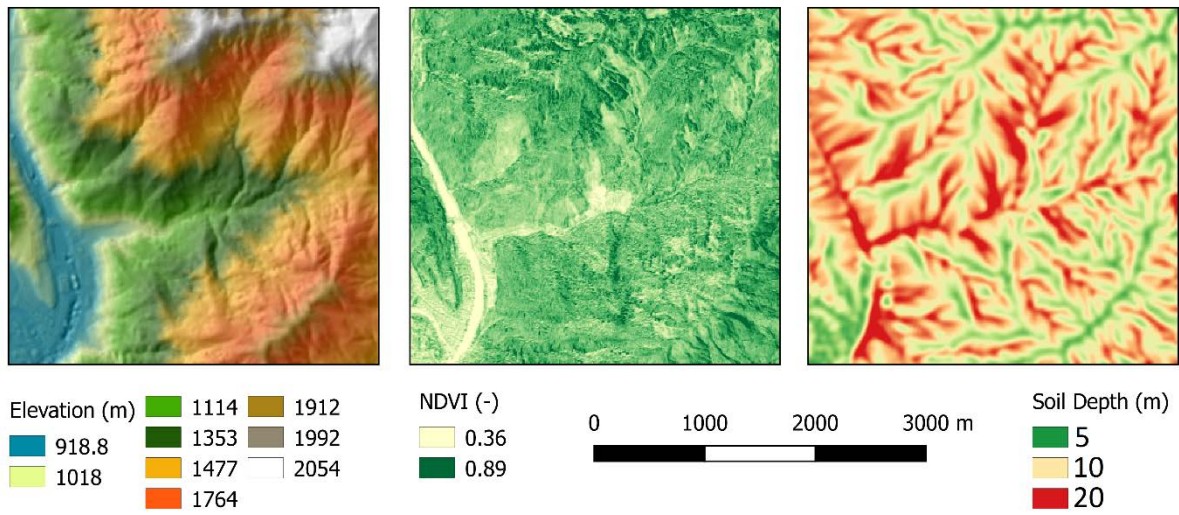


**Figure 6 An overview of the input data for the Hongchun catchment: elevation model (left), NDVI (middle), modelled soil depth (right).**

Spatial seismic acceleration data were obtained from the USGS ShakeMap product (Wald et al., 2005), developed using combinations of measurements and intensity prediction equations, and Peak Ground Acceleration (PGA) values were around

1.5 g for this earthquake in the Hongchun area. Landslide material strength parameters were obtained from tri-axial strength tests performed for engineering reports ordered by the local government (Table 2; Yang, 2010; Hao et al., 2011; Li et al., 2011). The consolidated-drained triaxial tests were performed on samples acquired from both the deposition and source areas, although exact locations are unknown. The resulting average values for cohesion of 7.3 kPa and internal friction angle of 27 degrees were calculated from 16 samples. While the cohesion and density matches the values found in other studies (Ouyang

et al., 2015; Domenech et al., 2019), the internal friction angle was estimated to be between 30 and 40 degrees in a previous study (Ouyang et al., 2015), based on measurements of internal friction angle on similar igneous rocks in the nearby Xiaojigou ravine. The difference between these values can be explained by the material used in the tri-axial tests. In our case, material from landslide depositional areas or areas close to a landslide source was used. These samples contain more fractured and loosened material. Thus, these measurements will provide a more appropriate value in post-earthquake entrainment

simulations. However, for pre-earthquake slope stability estimations, a value of 35 degrees was used, as was done in other studies (Ouyang et al., 2015; Domenech et al., 2019). Textures were found to be clay-loam with large amounts of gravel.



Saturated infiltration rates were measured in the field by using field ring infiltration tests. Because of the large amounts of gravel, and macro-pores within the materials structure, infiltration values were relatively high. Values were obtained for a total of four locations, which gave an average value of 65 mm/h for saturated conductivity. Other soil related parameter such as

porosity, density and matric suction were obtained using the pseudo transfer functions from Saxton et al. (2006).

**Table 2 Strength parameters for the debris flow material in the Hongchun catchment (Yang, 2010; Hao et al., 2011; Li et al., 2011) and saturated hydraulic conductivity.**

|  | Average (16 samples) |
|---|---|
| Cohesion (kPa) | 7.3 |
| Internal Friction angle (Degrees) | 27.0 |
|  | Average (16 samples) |
| Density (kg/m3) | 2145 |
| Median Grain Size (mm) | 3.1 |
|  | Average (4 samples) |
| Saturated Hydraulic Conductivity (mm/h) | 65 mm/h |

Soil depth values were obtained by applying the spatial soil depth model from Ruette et al. (2013), which uses steady-state assumptions to balance material production with soil movement according to an empirical formulation of erosion and lateral

transport. The production of weathered material depends on the weathering rate of bedrock. Since weathering rates are generally difficult to obtain, this parameter becomes the main calibration parameter. Validation was done using soil depth values obtained from landslide scarps. In this study, we used the pre- and post- earthquake elevation model differences provided by Tang et al. (2019). From their data, we sampled elevation model differences in landslide source areas to obtain failure depths, assuming that the failures were at least as deep as the top layer of weathered material. In case of shallow landslides,



we took the maximum landslide depth as soil depth. For the larger, deep-seated landslides, no samples were taken since it was

not possible to obtain a good estimate of the depth of the top layer. Additionally, we simulated a second layer within the slope

stability calculations with an additional depth of 20 meters. We assumed that this second layer did not contain groundwater,

had an internal friction angle of 35 degrees, was subject to sub-surface lateral forcing, and influenced by the weight of the

upper layer. Calibration results and cumulative distributions of depth values are shown in figure 7.

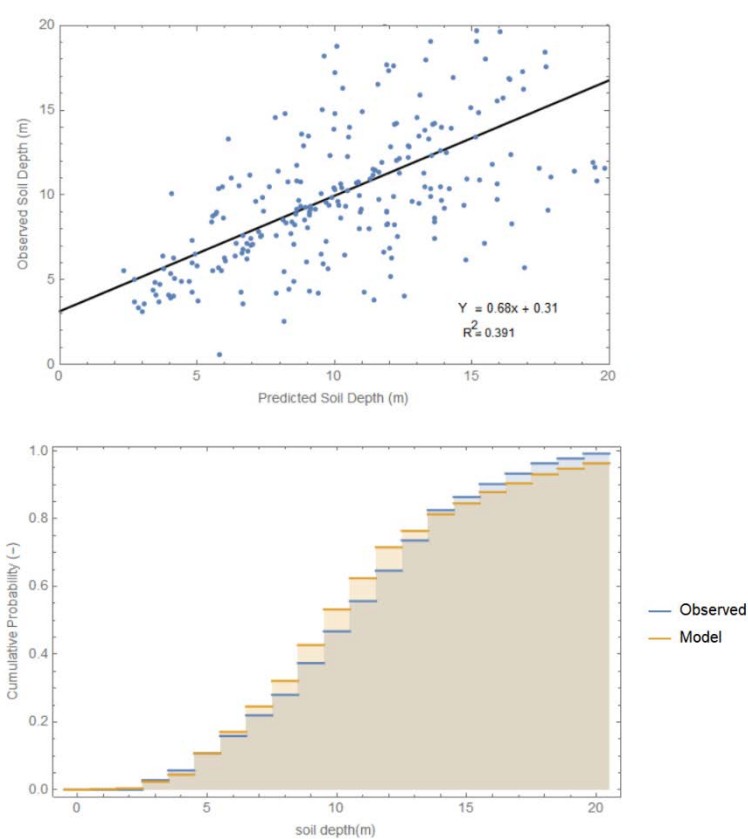


**Figure 7  Soil depth simulation results. (Left) A comparison of predicted vs observed values. (Right) Probability distribution for observed and simulated soil depth values.**

The effect of vegetation on slope stability and entrainment rates were taken into account by adding root cohesion. The recovery

of vegetation in the landslide affected areas developed through distinct stages (Huang et al., 2015). We estimated the average

root cohesion for the various vegetation classes using a combination of literature data and measurements (Schmidt et al., 2001;

De Baets et al., 2008; Chock et al., 2015). We used the method as described by De Baets et al. (2008) which requires measuring





root critical force and root diameters for all significant roots within a specific surface area. Then, after converting force to tensile strength, the total effect of the roots per unit area can be estimated using equation 40.

$$40 \qquad c_{root} = \frac{\sum T_i n_i a_i}{A}(\sin(\theta) + \cos(\theta)\tan(\phi))$$

Where $c_{root}$ is the added apparent root cohesive strength (kPa), $i$ is the root diameter class, $T_i$ is the root tensile strength (kPa), $n_i$ number of roots within the diameter class, $a_i$ is the cross-sectional area of the root, $A$ is the area of the soil occupied by roots (m2), $\theta$ is the angle of shear distortion in the shear zone (°) and $\phi$ is the internal friction angle (°). Note that term related to the shearing angle $(\sin(\theta) + \cos(\theta)\tan(\phi))$ is usually assumed to be approximately 1.2 (Baets et al.,2008). Here, we used a spatial calculation of this variable.

A total of 12 measurements of root cohesion averaged over a 0.1 m$^2$ area were done. On average, 108 roots were found per test site of 0.1 m$^2$, with an average diameter of 3.6 mm. Average values for young, post-landslide vegetation where found to be 4.8 kPa, averages for medium-sized vegetation where found to be 6.2 kPa. For mixed forest, literature values were used, and root cohesion was estimated to have a value of 8 kPa (Chock et al., 2015; Huang et al., 2015). The estimated cohesion values were combined with the land cover map in order to predict root cohesion spatially. Finally, within each land cover class, 445 we linearly scaled the root cohesion to fractional vegetation cover values that were derived from the NDVI both before and after the earthquake.

### 3.3 Calibration and Validation

The predicted co-seismic failure areas were calibrated using the landslide inventory and mapped deposition based on aerial imagery and pre-and post- event elevation data differences. Within the inventory, no separation between source and deposit 450 areas was provided. Therefore we assumed that, based on field visits, on average the 25 % highest part of each landslide polygon reprented the source area. In order to compare the outputs of our model with others, we parameterized two other regional slope stability models. The first of these is Scoops3D, which uses random spheroid sampling to find the landslides with the lowest- failure volumes, and lowest factors of safety for each pixel (Reid et al., 2015). This model can also use seismic shake maps as input. The second model is r.slope.stability, which uses random ellipsoid sampling to find the lowest factor of 455 safety and failure depth for each pixel (Mergili et al., 2014). This model cannot incorporate seismic acceleration, so the output is calibrated without addition of a seismic forcing. The Newmark displacement method (Newmark, 1965) was not used in our



comparison. The primary reasons were the lack of predicted failure depths and the high similarity to the infinite slope model. When the critical acceleration is taken as threshold and failures occur when accelerations go beyond this value, results are identical to an infinite slope model that incorporates seismic acceleration. For both Scoops3D and r.slope.stability, the

subsurface description is identical as to the input for OpenLISEM Hazard. As a measure of model fit, we used the Cohens Kappa value. This metric shows benefit over simple accuracy, especially for modelling landslide occurrence, since it compensates for the large amounts of true negative predictions (no landslide in model and inventory) that usually dominate landslide study sites (Bout et al., 2018).

The second phase of the modelling first simulates hydrology and a debris flow initiated by entrainment. Then, within the same

simulation, the Min River is blocked by debris flow material, and Yinxiu town is flooded. Calibration for this part of the modelling is based on mapped deposition extent in the Min River, the flood extent in Yinxiu Town, and the estimated depositional volume (approximately $7.11 \times 10^5 \ m^3$).

Calibration of the entrainment and debris flow runout was based on the final spatial deposit extent at the catchment outlet. The calibration parameters for each part of the simulation are shown in table 3. An exception to this was the entrainment constant,

for which no clear guideline was known to determine the value for specific terrain types. Based on earlier simulations and flume tests from Takahashi et al. (1992), a starting value was chosen of 0.05. For initial soil moisture content, the value is cut off at full saturation. Initially, each parameter was varied by choosing values between 60 and 140 percent of the original value. After initial calibration was done, the parameters were adjusted according to the steepest descent principle, in order to find the best set of parameter values. To have a level of validation in the simulations, the parameters resulting from calibration of the

first chain were used as input for the second chain.

**Table 3 Calibration parameters, their initial values and their final calibrated values for both chains.**

| Simulation Part | Calibration Parameters | | |
| --- | --- | --- | --- |
| Co-Seismic Slope Failure | Soil Depth | Internal Friction Angle | Soil Cohesion |
| Original Average | 4.5 | 27 | 7.3 |
| Calibrated Multiplier | 1.3 | 0.91 | 1.2 |



| Hydrology, Flow and Entrainment | Entrainment Constant | Initial Soil Moisture Content | Manning's N |
|---|---|---|---|
| Original Average | 0.005 | 80 % | 0.127 |
| Calibrated Multiplier | 0.6 | 1.12 | 0.82 |

## 4 Results

Simulation results for the first chain, co-seismic slope failure and runout, are shown in figure 8 and 9. Both the accuracies and

Cohens Kappa values for each of the used slope stability models are shown in table 8.

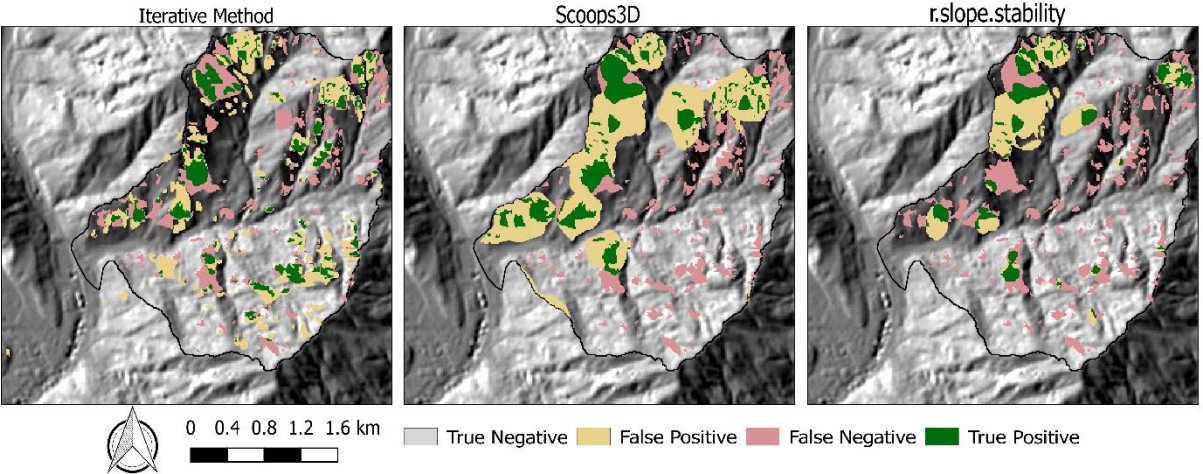

**Figure 8 A comparison of simulated slope failure extent with mapped co-seismic slope failures. (Left) OpenLISEM Hazard Iterative Failure with sub-surface forcing, (Middle) Scoops3D random spheroid sampling. (Right) r.slope.stability random ellipsoid sampling.**

**Table 4 Slope stability simulation accuracy and Cohens Kappa values.**

| Model | OpenLISEM Hazard | Scoops3D | r.slope.stability | Mapped |
|---|---|---|---|---|
| True Negative (m$^2$) | 3775700 | 3420700 | 3971800 | 4258397 |
| False Positive (m$^2$) | 530100 | 476100 | 627800 | 0 |





| False Negative (m$^2$) | 587300 | 939300 | 391200 | 0 |
|---|---|---|---|---|
| True Positive (m$^2$) | 315400 | 369400 | 217700 | 950103 |
| Accuracy | 79 | 72 | 81 | 100 |
| Cohens Kappa | 0.232 | 0.181 | 0.190 | 1 |
| Average Size (m$^2$) | 36394 | 140253 | 94028 | 21547 |


The general spatial patterns are predicted reasonably by all models, but the models differ considerably in details. For the ellipsoid and spheroid sampling done by r.slope.stability and Scoops3D, the failures are larger than those actually mapped (Table 4). For the iterative method, sizes are mixed but much more similar. Major landslides, in particular in the north, are predicted with similar size, although not on the exact location, by all models. The highest accuracy (81 %) is obtained using
r.slope.stability, and the highest Cohens Kappa (0.232) value with OpenLISEM Hazard. The OpenLISEM Hazard iterative method shows the best reproduction of the general pattern, in particular since both Scoops3D and r.slope.stability lack slope failures in the southern half of the catchment. Of the three models, r.slope.stabiltiy and OpenLISEM Hazard show similarity in total failure area when compared to the inventory. However, the results are not accurate enough for any reliable prediction of future events.


**4.1 Runout and the blocking of the Hongchun stream**

When slope failures are simulated, OpenLISEM Hazard automatically introduces landslide runout by transferring the failed volume and its properties to the Mohr-Coulomb solid-fluid mixture flow equations. The depth of slope failures determine directly the amount of solids and fluids introduced (Figure 12D). The landslide material moved down the slopes into the main
channel of the Hongchun watershed, blocking it in at least one location (Figure 9A). The accuracy for the calibrated simulation was 64 percent with a Cohens Kappa value of 0.28 (Table 5), which is mainly be attributed to the accuracy of the failures that started this process (Figure 9B). Runout distances are similar to those mapped, indicated by the landslides reaching the main channels within the Hongchun catchment, but not reaching the Min river. In one location the channel was blocked by a deposit of 12 to 18 meter deep which was simulated with high accuracy as compared to the mapped blockage (Figure 9B). Engineering



reports indicate a similar depth (16 meter) for the landslide dam (Yang et al., 2010; Hao et al., 2011; Li et al., 2011). We

considered to use the landslide initiation polygons from the inventory to increase the accuracy of the runout simulation.

However, since the aim of this research is to provide a true multi-stage modelling setup, we used the integrated prediction of

slope failures as input in the runout modelling.



**Figure 9 (A) Maximum landslide runout flow depth. (B) The simulated final deposit depth of the landslides. (C) A comparison of modelled landslide runout with the mapped landslide inventory. (D) Initiation depth from the slope failure simulation.**



**Table 5. Confusion matrix for the landslide runout prediction in Hongchun watershed.**

| Model | Runout $(m^2)$ |
|---|---|
| True Negative | 2092900 |
| False Positive | 1260500 |
| False Negative | 613900 |
| True Positive | 1241200 |
| Accuracy | 64 |
| Cohens Kappa | 0.28 |


### 4.2 Validation of failure and runout for the major central landslide using elevation model differences

Two large landslides occurred in the northern part of Hongchun catchment. One of those, near the center of the area, blocked the main channel of the catchment. For this landslide, pre- and post-earthquake elevation models from LiDAR data are available (Tang et al., 2019), which were used to calculate the landslide volume. For this particular area, we compared the

predicted elevation model differences based on slope failure and deposition from our modelling chain with the results from the LIDAR DEMs (Figure 10).

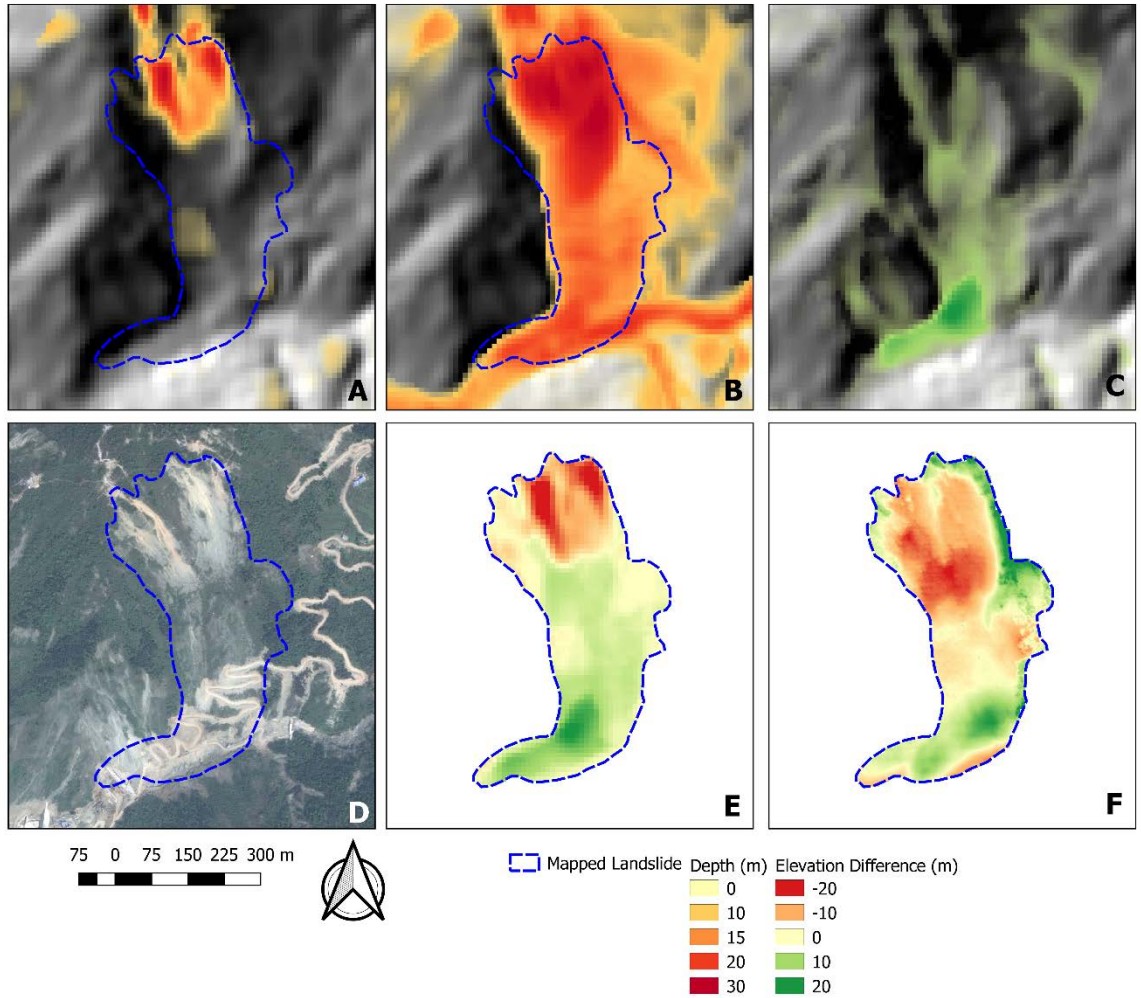

**Figure 10 An overview of the central largest landslide in the Hongchun watershed. (A) The simulated failure depth, (B) The simulated maximum runout depth, (C) The simulated deposition depth, (D) Post-Earthquake satellite image (Worldview, 2011) Note the mining activities in the landslide deposit area (E) Predicted elevation model differences due to co-seismic landslides. (F) Observed elevation model differences from pre-and post-earthquake LiDAR data.**

Both the slope failure and runout thicknesses are predicted with high accuracy for this landslide. The landslide deposits of around 20 meters thick remained in the main channel without spreading significantly. These deposits have later been mined as materials for local construction. As can be seen in Figure 14 the failure, deposition and elevation differences are highly similar with matching spatial patterns.



### 4.3 Simulation of the second multi-hazard chain

The results from the modelling of the second stage show a full physically-based simulation that reproduces the behavior and the impact of the event (Figure 11). This rainfall event with two distinct peaks resulted in a rainfall amount of 220 mm in two days, which was modelled in time steps of 0.5 seconds. Due to the large rainfall volume, runoff increased rapidly leading to

large amounts of sediments that were entrained from the co-seismic landslide deposits. Within the simulation, entrainment takes place after runoff has converted into streams. There, higher water pressures and velocities provide shear stress on the surface that is sufficient to overcome the materials internal stability. Near the outlet of the watershed, entrainment decreased due to a decrease in slope steepness. This decreased flow velocities, but also increased internal stability of the available material. Because of this, entrainment stopped, the flowing material lost momentum and was finally deposited in the Min

River.

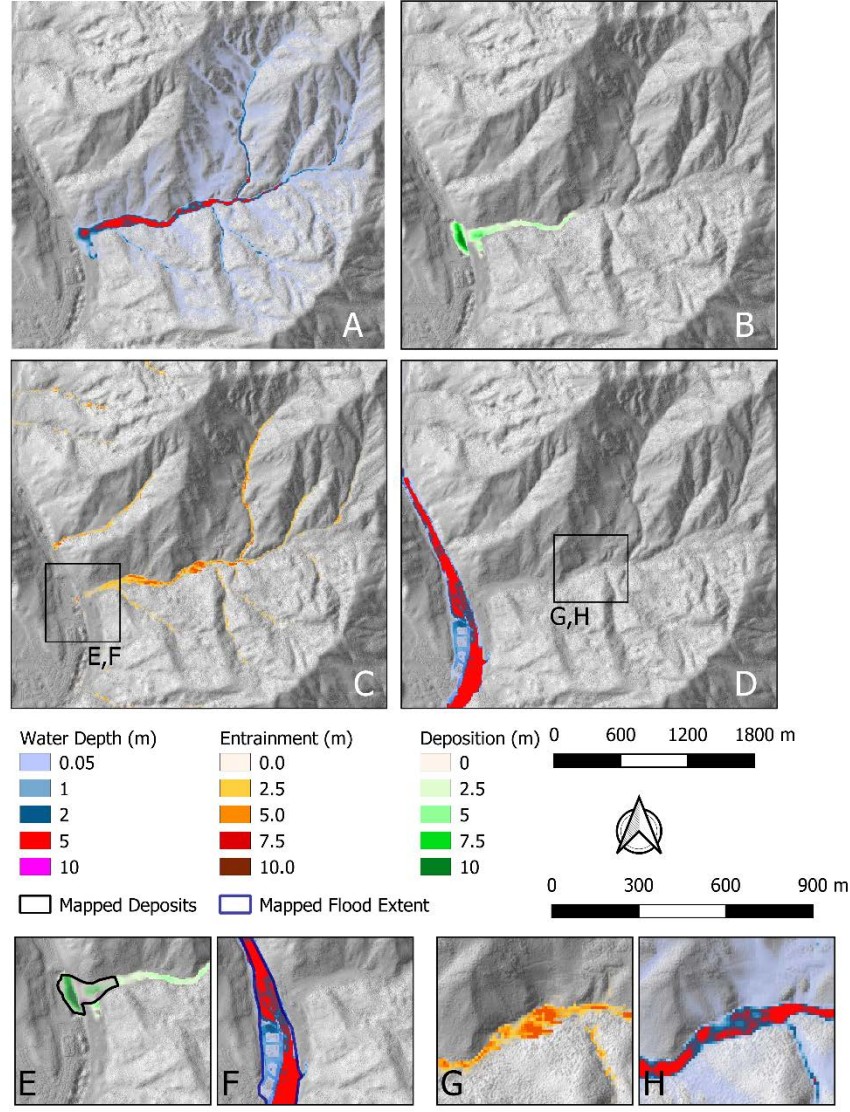

**Figure 11 Calibrated simulation results for the second chain in the Hongchun watershed. (A) Maximum total flow depth; (B) Final deposit depth; (C) Entrainment depth; (D) River flood depth; ( E & F): Zoom of Hongchun outlet with (E): Deposition depth compared with mapped extent, (F) River flood depth compared to mapped flood extent; (G &H) Zoom of Hongchun Landslide Dam with (G) Entrainment depth and (H)Maximum flow depth.**

*Table 6 Confusion matrix, accuracy and Cohens Kappa values for the debris flow deposition and flooding of the Min River.*

| Model aspect | Deposition | Flood |
| --- | --- | --- |



| | | |
|---|---|---|
| True Negative (m²) | 516000 | 331543 |
| False Positive (m²) | 28000 | 34510 |
| False Negative (m²) | 35000 | 51890 |
| True Positive (m²) | 213000 | 546132 |
| Accuracy | 92 | 91 |
| Cohens Kappa | 0.84 | 0.81 |

The simulated debris flow reaches the Min River and the deposits accumulate in the river. Most of the momentum has been lost before the material leaves the Hongchun watershed, and even though the Min River has a velocity of 7 m/s the deposited

volume is too large to be eroded by the Min river. Both the total deposit volume in the river, and the extend of the deposits compare well to the measured and mapped values. The deposits were modelled with an accuracy of 92 % and a Cohens Kappa value of 0.84 (Table 6). The modelled deposited volume is $5.82 \times 10^5 \ m^3$ (the observed estimation of the deposited volume is $7.11 \times 10^5 \ m^3$)

As a final stage of the event, the model predicted the flood behavior in the town of Yinxiu by the Min River, as its main course

was blocked by the debris flow deposits. The modelled damming area correlates well with the observed deposit volumes and flood extents, obtained from the interpretation of aerial images and field photos from the event (Figure 14). The flood accuracy is 91 % with a Cohens Kappa value of 0.81 (Table 6). Since the elevation data was up scaled from 2 meters resolution to 10 meters, the multi-story buildings merged with small streets to become a joined obstacle for the flow, which corresponds to the visual observations.

While validation using comparison to mapped extent is a useful tool, it does not guarantee all aspects of the hazard are correctly simulated (e.g. timing, velocities, heights). The modelled timing of the debris flow (between 03:30 and 06:00) matched the reported of 03:30 (Figure 12).

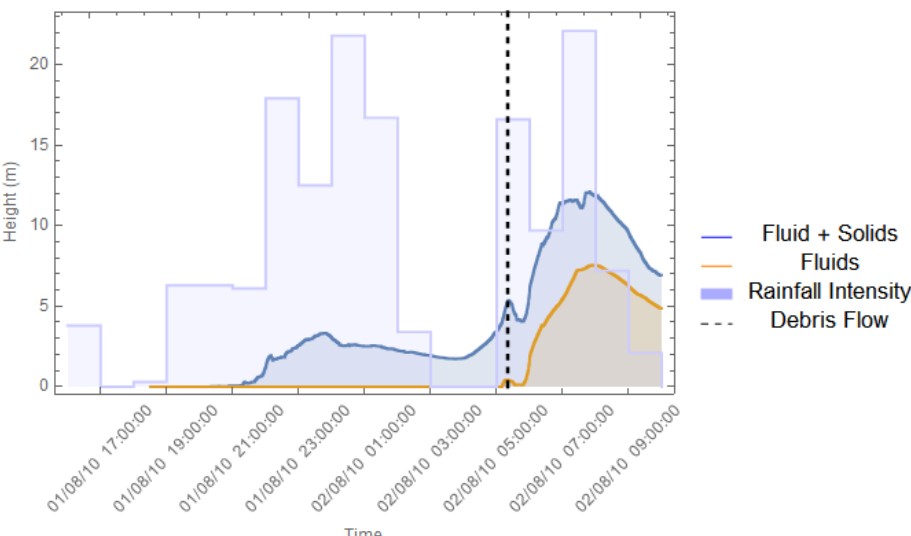

*Figure 12Time series data for rainfall, total flow height and solid flow height at the Hongchun outlet. Reported debris flow*

*occurrence time is indicated as 'debris flow'.*

# 5 Discussion

## 5.1 Uncertainties in modeling the multi-hazard chains

Several major obstacles in the simulation of co-seismic landslide occurrence can be identified, which are related to the

assumptions and techniques used in the models . The iterative method that is implemented in OpenLISEM Hazard, assumes

that, at least initially, failure surfaces are parallel to the terrain surface. This assumption can lead to a variety of issues when

the terrain has small-scale variations that do not represent the overall topography. This assumption is not present in the random

ellipsoid sampling methods. The iterative method shows similar failure depths and locations, but generally separates slope

failures more, where the other models show larger joined failures. The predominant cause of this behavior most likely lies in

the sub-surface force iteration, where the potential failure surface is also initially assumed parallel to the surface. Then, when

an obstacle exists that alters slope around several pixels, the sub-surface forces are blocked, and potentially larger failures

become disconnected (Figure 13).





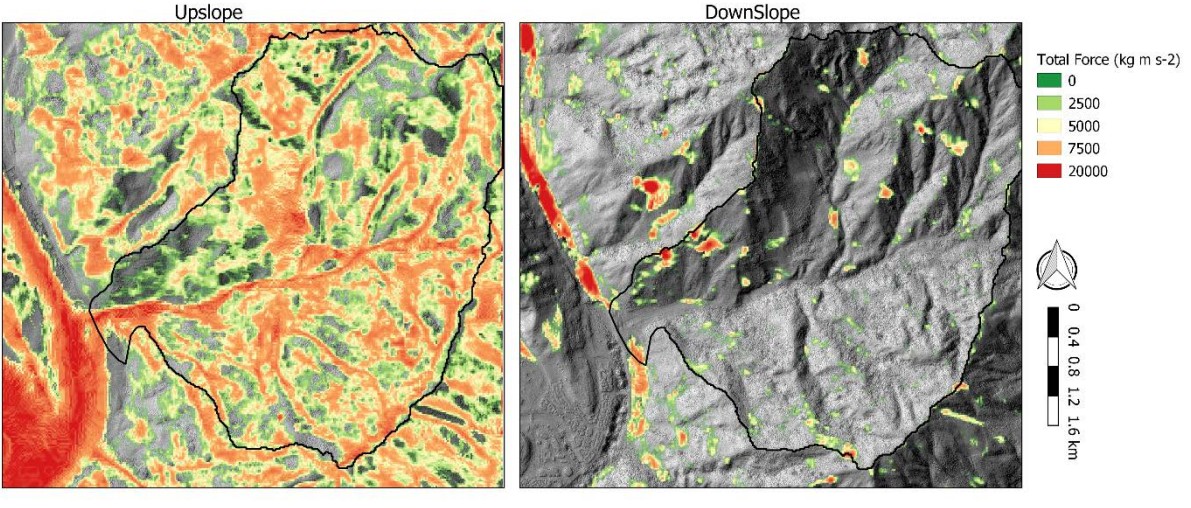

*Figure 13 The upslope (left) and downslope (right) additional forcing that is estimated based on an iterative solution for sub-*

*surface force redistribution.*

Secondly, structural geological input data are not available, and cannot be used in the iterative method. For three-dimensional analysis using random ellipsoid sampling, structural weaknesses could be implemented by altering strength properties of specific layers. When such data are available, applications of the detailed random ellipsoid sampling method, can allow for a greater predictive value (Mergili et al., 2014; Cance et al., 2017; Tun et al., 2018).

Furthermore, seismic acceleration of material is a complex and dynamic process, driven by seismic waves compressing and stretching the sloping materials. Such waves reflect and refract based on material properties and boundary conditions, which leads to a variety of topography or material-based amplifications. These amplifications are generally important in landslide hazard modelling (Jafarzadeh et al., 2015). However, shake maps produced by the USGS utilize empirical predictive functions to spatially extend ground accelerations, and ignore these crucial local amplification effecta. There is therefore a high spatial

uncertainty in the peak ground acceleration values we have used as input. Such uncertainties can be overcome by linking slope stability approaches to seismic wave modelling, a computationally heavy task that has of yet not been performed for the Wenchuan earthquake.

The final calibrated values of the input parameters were within a reasonable range (between 50 and 150 percent ) with respect to the original estimated ones. Two parameters stood out significantly in the calibration process. The first of these, soil depth

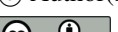



is a crucial parameter, as it determines the amount of material that is released, which is of direct influence in the runout

modelling. This can indirectly influence the amount of material available for entrainment by flow at a later stage. In an active

landscape, soil depth patterns are determined by rock weathering rates and mass transport/wasting. Soil depth increases until

there is sufficient material to induce slope failures or significant erosion. As a results, at many locations, combinations of soil

depth and slope can be near a critical state. The spatial component of soil depth is therefore of high importance. Despite the

importance, soil depth is very difficult to measure over larger areas, and has a high uncertainty in most model applications

(Kuriakose et al., 2009). Because information on soil depth is difficult to obtain through direct observation or remote sensing,

it is generally obtained through modelling approaches (Kuriakose et al., 2009; Ruette et al., 2013). Another reason for the

uncertainty in soil depth information is that a soil layer is a theoretical concept that does not always translate well into reality,

where weathering is a more gradual process. A second variable which was difficult to estimate is the entrainment constant.

Currently, there are several types of entrainment equations available in the literature, but most of these lack significant

guidelines for selecting practical entrainment parameters (Iverson and Ouyang, 2015). Besides this coefficient, other

parameters used in the entrainment equations, such as soil cohesion, internal friction angle and moisture content were measured

in the laboratory and are common values for geotechnical research. However, the number of samples that are tested for these

geotechnical parameters are always limited, and their spatial variability is generally high.

**5.2 Ensemble Simulations**

To analyze the uncertainties within the multi-hazard multi-stage modelling setup, we extended the calibration process to an

ensemble analysis. In total, for each of the 6 calibration parameters, 3 equal-interval values were used (calibrated values, and

the values plus or minus a given range of 10 – 50 %, as indicated in table 7. Thus, the first simulation was repeated 27 times.

The best estimate was used as input for the second simulation, which was similarly repeated 27 times. In total, 56 simulations

were thus performed, out of which normalized frequencies were generated for hazard occurrence. We defined a threshold of

0.25 meters above which a flow is counted as an actual hazard occurrence to avoid that the results would include runoff with

insignificant depths.

*Table 7 Parameter settings for the ensemble simulations.*



| Co-Seismic Slope Failure | Soil Depth (m) | Internal Friction Angle (radians) | Soil Cohesion (kPa) |
|---|---|---|---|
| Calibrated Value (average) | 5.85 | 24.57 | 8.8 |
| Variation range (+/- %) | 30 % | 30 % | 30 % |
| Hydrology, Flow and Entrainment | Entrainment Constant | Initial Soil Moisture Content | Manning's N |
| Calibrated Value (average) | 0.003 | 89% | 0.10 |
| Variation range (+/- %) | 50 % | 10 % | 20 % |



Figure 14 shows an ensemble plot for each stage of the simulation. For each location, these maps show the normalized

probability of hazard occurrence within the ensemble of simulations.

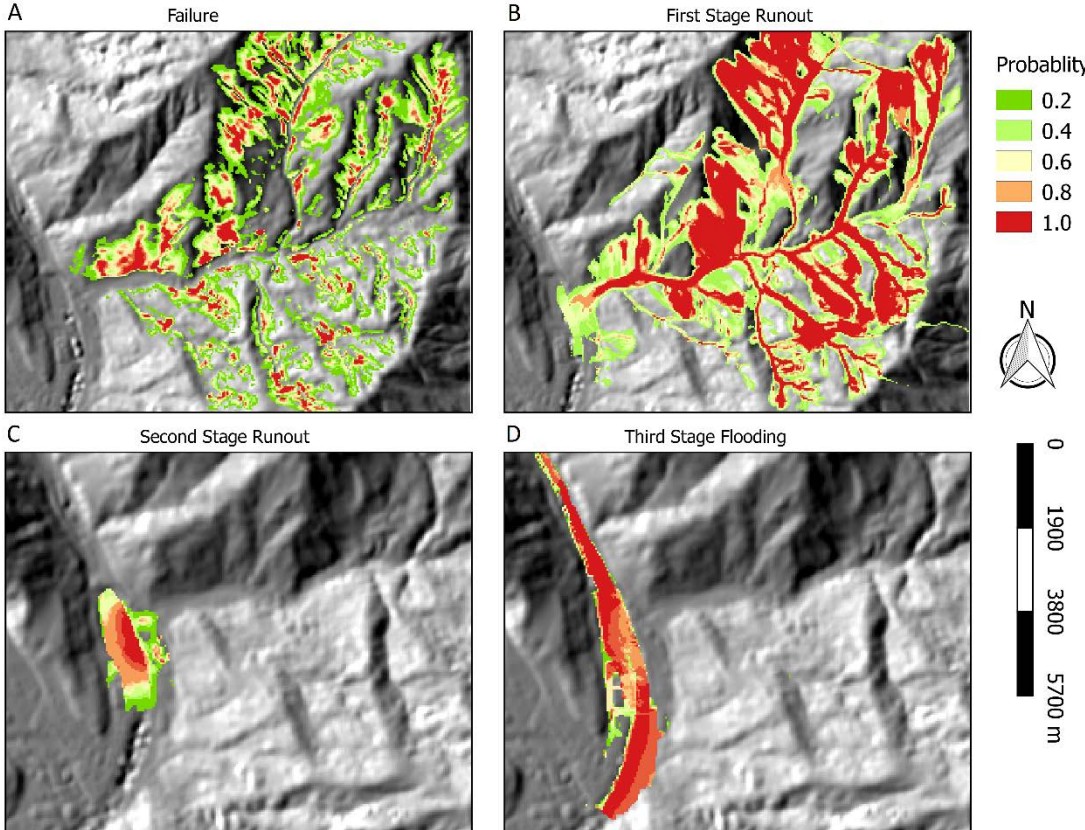

*Figure 14 Ensemble simulation results for the Hongchun watershed. Visualized is the normalized probability, based on the*

*ensemble of runs with varying input parameters, of the hazard occurring at each location. (A) Co-seismic slope failure. (B)*

*co-seismic landslide runout. (C) Post-seismic debris flow deposition. (D) Post-seismic river flooding due to blockage.*

The four maps in Figure 17 show illustrate how much the variation of the input parameters influences the simulated hazard for

the four stages of the multi-hazard chain. The prediction of co-seismic slope failures shows the highest spread, and therefore

the highest influence on the total event variability. The uncertainty in the input parameters influences the slope failure equations

most, since we have simulated the multi-stage event as an integrated sequence where the uncertainties in one process influence

various other processes. Despite the uncertainties, there is a substantial certainty that flooding of Yingxiu town will occur,



independent of the input parameters. For all performed simulations, the deposition volume in the Min river is never below approximately a third ($2.1 \times 10^5 \ m^3$) of the estimated volume ($7.11 \times 10^5 \ m^3$). For at least 80 percent of the simulations, there is at least some flooding experienced with a depth above 50 centimeters in Yinxiu. In comparison to other multi-stage hazard studies, these numbers are relatively high (Mergili et al., 2018b). Mergili et al. (2018b) reported that threshold behavior

and non-linear effects dominated the alterations in model behavior with changing parameters. In this current study, threshold effects most likely exist related to the landslide dam behavior in both the Hongchun watershed and the Min river. With altered input, there could be insufficient material for landslide dam creation, or not enough flow and entrainment for breaching. However, we do not encounter scenarios that fall below the threshold of breaching in our ensemble.

There are several factors that drive the high likelihood of flooding in Yinxiu town. The event is initiated by several triggering

conditions, or *forcings*, such as the seismic acceleration and rainfall, which occur in a very steep watershed, which determine the behavior of the hazardous processes. Seismic accelerations during the 2008 earthquake reached the 10[th] and highest USGS category for seismic acceleration (>139% g). Similarly, the 2010 rainfall event resulted in 220 mm of accumulated rainfall within 48 hours, with significant rainfall in the weeks before. The prediction of hazardous process caused by less extreme triggering events, for example the occurrence of landslides long duration by low intensity rainfall, is a much larger challenge

for physically-based modelling (van Beek, 2002). In our case, material strength parameters must deviate significantly from their measured and estimated values in order not to result in slope failure under such extreme seismic acceleration. Thus, the severity of the triggering events causes the simulations to predominantly show major slope failures, debris flows, deposition in the Min river and finally flooding of Yinxiu town.

### 5.3 Usability

A major challenge for the application of such type of multi-hazard modelling is the step from back-analyzing past events to forward prediction of future events. In the case of back-analysis, calibration is possible on data from the event itself, whereas predicting future events must be done without knowledge of the hazard that will occur. By calibrating on data of multiple events, and validating on other events (checking for accuracy using calibrated input from other events), a more robust model can be established. When validated on similar events, a higher level of confidence can be generated in the abilities of the model

to predict future events. In the case of the Hongchun watershed, the accuracy of the presented numerical experiments depended

on thorough calibration. This is caused by several factors, such as the uncertainties in the input data and the limited accuracy of the assumptions made within the model equations. In studies where multi-stage multi-hazard events are considered, this becomes even more difficult due to threshold effects and stacking uncertainties (Mergili et al., 2018). Despite these issues, calibration is possible by performing model runs with a large set of varying input parameters and comparing the output with measured data on the event (e.g. discharge, extent of landslides, debris flows or floods). Validation options, on the other hand, may be very limited as similar events are very rare. Debris flows occur more frequently within the watershed and can therefore be both calibrated and validated (Domenech et al., 2019). However, the multi-hazard chain as described in this research is a rater unique sequence of events that does not easily allow for validation, except when looking at individual components. Even then the co-seismic slope failure and runout stages do not occur frequently within the same region. Thus, we run into two primary issues in the predictive use of multi-stage and multi-hazard modelling. Firstly, the complex modelling has internal uncertainties that propagate, stack and provide a challenge in calibration. Secondly, such events occur with very low frequency (Kappes et al., 2012), and can therefore rarely be further calibrated and validated. Despite the difficulties in the calibration and validation process for multi-hazard multi-stage modelling, we have shown that for the Hongchun watershed, many aspects of the event sequence had a high likelihood despite the uncertainties. This indicates the possible use in predicting future events, especially when ensemble analysis is performed to show the relative spread of model results. However, this is computationally intensive, and cannot be carried out over large areas yet.

## 6 Conclusions

A physically-based model is presented that allows for the simulation of multi-stage and multi-hazard chain events in mountainous terrain. Catchment-based hydrology was combined with slope stability analysis under seismic acceleration. Entrainment of loose material was implemented to alter both topography and the composition of the flow. The modelling setup is able to fully simulate the behavior and impact of multi-stage and multi-hazard events. The developed model code is available as part of the on-going development of the open-source OpenLISEM Hazard model (https://sourceforge.net/projects/lisem/). The model was tested in the Hongchun watershed, located near the epicenter of the 2008 Wenchuan earthquake, in two sequences of hazard interactions. The first sequences was the generation of earthquake induced landslides, that resulted in



runout of landslide materials and the blockage of a stream channel. The second sequence was the triggering of debris flows by extreme rainfall, resulting in the breaking of the existing landslide dam, and the formation of a debris flow fan in the main river, forcing the river to flood the city on it banks. The modelled slope failures show a reasonable accuracy when compared with the actual landslide inventory. The failure simulation is the least accurate process within the simulation (accuracy 64 %).

Improvements in this process might be possible using a higher quality ground shaking input map that includes topographic amplification and other non-linear terrain-induced effects. Furthermore, a better spatial estimate of the sub-surface structure and the three-dimensional variability of strength parameters could improve the accuracy. However, current techniques for simulation of regional failure volumes are still under development (Wasowski et al., 2011). Over time , such type of modelling should be able to translate the modelling of seismic acceleration for many earthquake scenarios, into a probabilistic earthquake-

induced landslide hazard map.

The runout of the co-seismic landslides and the debris flow that was triggered two years later, were were predicted with high accuracy. Pre- and post- earthquake elevation model differences were used to validate the modelling outcomes spatially for the largest central landslide, where high accuracy elevations models were available for both pre- and post-earthquake periods. In particular, the blocking of the Hongchun stream by co-seismic landslide runout was simulated with good agreement

(accuracy 91 %). During the simulation of the 2010 rainfall event, a debris flow was initiated by entrainment, and the landslide dam was breached. Eventually, the model simulated with relatively high accuracy the flood behavior in the second stage of the event (accuracy 92 %).

The sensitivity of the model was further analyzed by looking at ensemble simulation results. Uncertainty propagated through the different stages of the simulations. Despite this uncertainty, the ensemble models showed a high likelihood of flooding in

Yinxiu. The model results show high certainty due to both the magnitude of the seismic accelerations, rainfall intensities, and the equations guiding the gravitational flows. This does not mean that the model is able to predict future event with equal reliability. However, it does indicate that, when the inherent uncertainties are taken into account, predictions from advanced spatially distributed and physically-based models can provide insight and have some predictive value. Simulation results should be treated with caution. However, with validation and advancements in the understanding of the physical processes,

multi-stage multi-hazard modelling might become a useful and trusted tool for decision makers in multi-hazard risk

assessment.

**Acknowledgements**

This research was supported by National Key Research and Development Program of China (2017YFC1501004) and National

Natural Science Foundation of China (41672299). We would like to acknowledge the help of State Key Laboratory of

Geohazard Prevention and Geo-environment. In particular, the assistance and advice from Dr. C. X. Tang, Dr. T. van Ash and

Dr. X. Fan was used in improving the quality of this research.

**Author Contributions**

All authors contributed significantly to this final works. Authors C. Tang and C. Tang carried out field works and laboratory

test fort he model, as well as helped develop the model theory. V. Jetten and C. van Westen provided assistance to the

hydrological modelling, interpretation/discussion of results as well as writing. B. van den Bout developed the model code,

carried out the simulations, and wrote the methodology.

**Code availability**

The LISEM model, which was used in this research, is an open-source multi-hazard modelling tools. The software and source

code are available from www.lisemmodel.com, together with documentation and example datasets. The source is written in

c++ and compiles on windows and linux based operating systems. Visit the github repository to get started

(https://github.com/bastianvandenbout/LISEM)

**Data availability**

A datasets for Hongchun Gully, which was studied in this work, is provided on the LISEM website (www.lisemmodel.com).
For more information on how to download and use this datasets, visit the downloads page.

**Parameter List**

$|\vec{u}|_{cr}$ is the critical velocity for deposition ($m\ s^{-1}$)

$\overrightarrow{F_{lat}}$ is the vector of laterally acting forces ($kg\ m\ s^{-2}$)

$h_s$ is the depth of the failure plane ($m$)

$S_f$ is the surface friction term (-)





$S_f$ the momentum source term ($m\ s^{-2}$)

$U_T$ is the settling (or terminal) velocity of a solid grain ($m\ s^{-1}$)

$U_T$ the settling velocity ($m\ s^{-1}$)

$z_0$ is the lowest neighboring elevation (m)

$\alpha_s$ is the volumetric solid content of the flow (-)

$\rho_f$ is the density of the fluid ($kg\ m^{-3}$)

$\rho_s$ is the density of the solids ($kg\ m^{-3}$)

$\tau_c$ is the critical shear stress (Pa)

$C_1 = c,\ C_2 = ((\gamma - m\gamma_w)z + m\gamma_w z),\ C_3 = ((\gamma - m\gamma_w)z)$ are simplifying compound parameters (-)

$K_s$ is the saturated conductivity ($m\ s^{-1}$)

$\vec{S}$ is the normalized slope vector (-)

$T_i$ is the root tensile strength (MPa)

$V_d$ is the darcy flow velocity ($m\ s^{-1}$)

$a_i$ is the cross-sectional area of the root

$c'$ is the effective cohesion of the soil (kPa)

$c^b$ is the cohesion of the bed material (Pa)

$c_{root}$ is the added apparent root cohesive strength (kPa)

$d_{50}$ is the median grain size ($m$)

$f_{pot}$ is the potential infiltration rate ($m\ s^{-1}$)

$n_i$ number of roots within the diameter class

$n_{td}$ is the turbulent dispersive coefficient ($m^{\frac{1}{2}}s^{-1}$)

$u_b$ is the basal velocity ($m\ s^{-1}$)

$\alpha^b$ is the volumetric solid concentration of the bed material (-)

$\alpha_{eq}$ is the equilibrium volumetric solid concentration (-)


$\gamma_w$ is the density of water $(kg\ m^{-3})$

$\theta_i$ is the initial soil moisture content $(m^3\ m^{-3})$

$\theta_s$ is the porosity $(m^3\ m^{-3})$

$\rho_{eff}$ is the total effective density of the flow $(kg\ m^3)$

$\tau_y$ is the yield stress (Pa)

D is the deposition rate $(m\ s^{-1})$

d is the grain diameter $(m)$

d is the median grain diameter (-)

D the force demand (denominator of equation 3)

E Is the rate of change of the basal topography (erosion rate) $(m s^{-1})$

F and G are the scalar functions describing the flow velocity of solids and fluids respectively (-)

g the gravitational acceleration $(m\ s^{-2})$

h being the flow height $(m)$

$h$ is the hydraulic head (m)

I the infiltration $(m)$

M is an empirical parameter depending on the Reynolds number ($\approx 0.2$) (-).

n is the Manning's n friction coefficient $(s\ m^{-\frac{1}{3}})$.

R is the rainfall $(m)$

Re is the particle Reynolds number (-)

S is the friction term $(m\ s^{-2})$

u is the flow velocity $(m\ s^{-1})$

α is the first viscosity parameter (-)

β is the second viscosity parameter (-)

η is the dynamic viscosity of the fluid ()

τ is the shear stress (Pa)



$A$ is the area of the soil occupied by roots $(m^2)$

$A$ is the mobility of the interface (-)

$C$ is the force capacity (numerator of equation 3)

$F$ is the cumulative infiltrated water $(m)$

$FOS$ is the factor of safety (-)

$H$ is the height of the flow (m)

$K$ is the resistance parameter for laminar flow (-)

$L$ is the length scale of the flow (m)

$i$ is the root diameter class

$m$ is the fraction of the soil depth that is saturated from the basal boundary (-)

$p$ is the calibration factor for the critical velocity for deposition (-)

$z$ is the elevation of the top surface (m)

$\alpha$ is the peak horizontal earthquake acceleration $(m\ s^{-2})$

$\beta$ is the slope angle (-)

$\gamma$ is the density of the slope material $(kg\ m^{-3})$

$\eta$ is the viscosity $(kg\ s^{-1}\ m^{-1})$

$\theta$ is the angle of shear distrortion in the shear zone (°)

$\psi$ is the matric pressure at the wetting front $(h = \psi + Z)\ (m)$

$\phi$ is the internal friction angle (°)

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
