# Peer review of "Physically-Based Modelling of co-seismic Landslide, Debris Flow and Flood Cascade"

_Natural Hazards and Earth System Sciences, 2021_

## Author Response (AR1)

Changes made to the manuscript:

Line … (new line number): Change made

Eq. 21 and 22: Source terms renamed to Sx,f and Sy,f

236 (237): Renamed source term to correct terminology. Ss to Sf.

Figure 5: Photo added at suggestion of reviewer.

410 (418): "pedotransfer functions" (not "pseudo transfer functions"), I think

440ff (411): we explained our usage of the root cohesion within the model.

455 (463): r.slope.stability includes seismic forcing (Newmark and pseudostatic), but these functions were added very recently and have not yet been thoroughly evaluated.

Figure 9: Order or sub-figures changed according to reviewer comments

561 (575): Explain in some more detail the correspondence of modelled and observed timing.

Fig. 12: Legend clarified, separate axis for precipitation

589 (603): "effects" instead of "effecta"

598 (612) : "as a result"

599f (614): "Despite its importance …"

604 (618): entrainment coefficient?

Fig. 14: Nice figure, but (i) legends should show classes instead of values, and (ii) the colouring is not ideal, as the medium probabilities are much paler that the low probabilities, which gives a strange visual impression.

626 (640): "show illustrate": remove one of the two words, and: "Figure 14", not "Figure 17".

634 (649): Clarification according to reviewer comments

644 (660): adapted sentence

658 (674): Mergili et al. 2018 a or b?

680 (685): "first sequence"

683 (700): "on its banks"

688 (705): wording changed according to reviewer comments

693 (710): "elevation models"

713 (730): "Authors C. Tang and C. Tang": there is only one author named C. Tang provided in the list of authors.

723 (740): "A dataset"

65-66 (68)   Please add at least a reference supporting this statement. e.g. Baggio et al. (2021) simulated the process of debris flow initiation through bed erosion releasing an input hydrograph characterized by a solid concentration equal to the 10 % respect the total input volume. (Baggio, T., Mergili, M., & D'Agostino, V. (2021). Advances in the simulation of debris flow erosion: The case study of the Rio Gere (Italy) event of the 4th August 2017. Geomorphology, 381, 107664.)

99-100 (98) clarification of the objective of the study

339 (340) specified the amount of area with a slope over 30 degrees to better indicate type of terrain. In addition a photo is now available in figure 5.

345      Is there an estimation of the total volume of the moved material? this was already in the paper, see also the reply to the reviewer comments.

Fig.4    To improve the understanding of the process you may consider to add a dem of difference map involving the pre- and post-earthquake DTMs. We have clarified in the text in line 388.

351 (353)      satellite instead of "ssatellite"

356  (358)    Please provide also the value of the second highest peak in rainfall and the total duration of the storm event. Here you can directly refer to Figure 12 for the rainfall pattern and to Table 1 for the rainfall source.

358 (350)      Can you provide also the length and width of the debris dam?

Tab. 1   Dates of the elevation datasets provided as suggested by reviewer.

385 (391) Clarified that we resampled to 10 meter resolution and did the modelling at that resolution.

391 (396) Specified that the shown data refers to simulation stage 1.

499 (508) ALtered the referred figure number.

528 This was already in the manuscript, see also the reply to the reviewer comments.

529  (538)  Corrected figure number to 10.

Fig. 11   Adapted to the comments from the reviewer.

Fig. 12   Has been adapted to reviewer comments with a clearer legend and axis.

626 (640)      I think it is Figure 14. Please check.

691 (708)      "Were" is repeated two times.